# Metal-organic framework and inorganic glass composites

Louis Longley [1,5], Courtney Calahoo[2,5], René Limbach[2], Yang Xia[2], Joshua M. Tuffnell[1], Adam F. Sapnik[1], Michael F. Thorne [1], Dean S. Keeble[3], David A. Keen [4], Lothar Wondraczek [2] & Thomas D. Bennett [1]✉

Metal-organic framework (MOF) glasses have become a subject of interest as a distinct category of melt quenched glass, and have potential applications in areas such as ion transport and sensing. In this paper we show how MOF glasses can be combined with inorganic glasses in order to fabricate a new family of materials composed of both MOF and inorganic glass domains. We use an array of experimental techniques to propose the bonding between inorganic and MOF domains, and show that the composites produced are more mechanically pliant than the inorganic glass itself.

[1] Department of Materials Science and Metallurgy, University of Cambridge, Cambridge CB3 0FS, UK. [2] Otto Schott Institute of Materials Research, University of Jena, Fraunhoferstrasse 6, 07743 Jena, Germany. [3] Diamond Light Source Ltd., Diamond House, Harwell Campus, Didcot Oxfordshire OX11, 0DE, UK. [4] ISIS Facility, Rutherford Appleton Laboratory, Harwell Campus, Didcot Oxfordshire OX11, 0QX, UK. [5]These authors contributed equally: Louis Longley, Courtney Calahoo. ✉email: tdb35@cam.ac.uk

Metal-organic frameworks (MOFs) are a class of porous three-dimensional framework materials produced from the self-assembly of inorganic nodes and organic linkers[1]. High-throughput synthesis techniques have resulted in over 80,000 reported crystalline MOF structures, in a broad variety of network architectures[2,3]. Accessible pore volumes may exceed 5.02 cm$^3$ g$^{-1}$ [4], and thus compare favourably with both activated carbons and inorganic zeolites[5,6]. These high internal volumes have led to proposed applications for crystalline MOFs in, e.g., gas storage and separation[7,8], catalysis[9], water harvesting[10] and low-κ dielectric applications[11].

Research into MOFs has traditionally focused on the crystalline domain. There has however been a growing impetus towards the synthesis and characterisation of non-crystalline structures[12]. In particular, the zeolitic imidazolate frameworks (ZIFs) are a sub-family of MOFs of composition M(Im)$_2$, where M is a tetrahedrally coordinated divalent metal cation, typically Zn$^{2+}$ or Co$^{2+}$, and Im$^-$ is an imidazolate (C$_3$H$_2$N$_2{}^-$) derivative[13]. Several ZIFs have been shown to have an accessible liquid state, formed by heating the crystalline solids to ~450 °C[14].

Inorganic glasses typically consist of multiple components, which help prevent crystallisation through introducing geometric frustration[15]. These inorganic materials possess the chemical, thermal and mechanical robustness required to be a structural component for applications, involving mechanical cycling with large stresses. The high mutual solubility of inorganic glasses[16] allows easy production of new materials with properties intermediate between two end-members[17], e.g., borosilicates, mixed-alkali glasses and aluminosilicates. This ability to tune physical properties is highly advantageous in industrial and technological applications, for example, in order to achieve specific mechanical performance[18]. In similar ways, organic polymer glasses may also be 'blended' to produce new, homogeneous amorphous materials with intermediate, and industrially relevant properties[19]. We have previously applied similar methodologies to MOFs, showing that glass-forming ZIF systems can be mixed in the liquid state to produce a structure with sub-micrometre scale domains, where the glass transition temperature is tuneable between that of the end-members[20].

Motivated by the relative advantages of MOFs, and inorganic glass materials families, here we detail the fabrication and characterisation of an unconventional class of composite materials, containing domains of both inorganic- and MOF- glasses. Specifically, we exploit liquid phase mixing between a liquid MOF and an inorganic melt, in order to create a composite material, which incorporates the mechanical, thermal and chemical properties of inorganic glasses while maintaining the chemical versatility of the MOF component.

## Results

**Materials selection**. To maximise the available temperature region for composite synthesis ZIF-62, [Zn(Im)$_{1.75}$(bIm)$_{0.25}$] (Im = C$_3$H$_3$N$_2{}^-$ and bIm = C$_7$H$_5$N$_2{}^-$) (Fig. 1a), was chosen as the glass-forming MOF component. This is because of the large temperature range between the melting point, $T_m$ (~437 °C) and the onset of thermal decomposition (~600 °C)[21]. A key factor in the selection of the inorganic glass component is the possession of a glass transition temperature ($T_g$) close to the $T_m$ of ZIF-62. This is to enable good mixing between the two liquid phases and therefore promote formation of strong interfacial bonding between the components. Another important consideration is the avoidance of chemical reactions, leading to the decomposition of the organic linkers. We therefore selected the inorganic glass series, with composition $(1-x)([Na_2O]_z[P_2O_5])$-$x([AlO_{3/2}][AlF_3]_y)$ (Supplementary Table 1)[22], which possesses $T_g$ in the

range 310–450 °C, and because previous literature indicated the chemical compatibility of phosphate groups and imidazole in hybrid inorganic systems[23].

Phosphate glasses are a widely explored family of inorganic glasses owing to their low melting temperatures and biocompatibility[24]. The $(1-x)([Na_2O]_z[P_2O_5])$-$x([AlO_{3/2}][AlF_3]_y)$ glass is comprised of two major domains: (i) chains of phosphate tetrahedra connected through bridging P–O–P linkages, with some terminal non-bridging oxygens (NBOs) associated with sodium (P–O··Na$^+$), and (ii) islands of Al(OP)$_4$F$_2$ octahedra, which are strongly bonded to the phosphate chains through Al–O–P bridging bonds (Fig. 1b)[25]. Three $(1-x)$ $([Na_2O]_z[P_2O_5])$-$x([AlO_{3/2}][AlF_3]_y)$ compositions were prepared, and analysed by energy-dispersive spectroscopy (EDS) (Supplementary Fig. 1). The chemical compositions of these glasses are given in (Supplementary Table 1) and in accordance with their compositions the resultant glasses are referred to as the base, Na-deficient and Al-rich compositions.

Crystalline ZIF-62 was synthesised by a method adapted from the literature[26] (Supplementary Fig. 2). Then equal weights of ZIF-62 and each inorganic glass were ball-milled together. Consistent with previous literature on MOF blends and composites[27], the full name for these physical mixtures takes the form (ZIF-62)($(1-x)[Na_2O]_z[P_2O_5]$)-$x([AlO_{3/2}][AlF_3]_y)$ (50/50). We use the shortened naming convention (ZIF-62) (Inorganic Glass) (50/50) here, for readability and clarity.

**Thermal characterisation**. The three (ZIF-62)(Inorganic Glass) (50/50) samples were investigated using differential scanning calorimetry (DSC). Each of the samples in this series was heated above the melting endotherm of ZIF-62 to 450 °C and the mixtures were held for either 1 or 30 minutes at this temperature. Measurements on all samples were also made during a second heating ramp to 450 °C. The two different high temperature isothermal times were used in order to measure the effect of liquid phase mixing between the inorganic and the ZIF-62 on the structure of the resulting composite.

Samples of crystalline ZIF-62 were also subjected to the same heat treatments (i.e., holding for 1 and 30 minutes at 450 °C), in order to provide a point of comparison for the thermal behaviour of the composite samples. The initial heating curves of ZIF-62 samples showed a melting event, with an offset at ~434 °C. The second heating curve of these samples, i.e., after they were held at 450 °C for 1 or 30 minutes and then cooled, displayed clear glass transitions at 322 °C and 314 °C (Supplementary Fig. 3), respectively, which is consistent with literature data on ZIF-62 and the resultant glass (termed a$_g$ZIF-62)[21].

The (ZIF-62)(Al-rich)(50/50) sample heated for 1 minute at 450 °C displayed melting of ZIF-62 ($T_m$(ZIF)), at 435 °C. This was followed by a rise in the baseline at ~440 °C (Fig. 2a), which was assigned to the glass transition of the inorganic glass ($T_g$(Al-rich)) by comparing with a DSC scan of the pure Al-rich glass sample (Supplementary Fig. 4). The second upscan showed two glass transitions; one assigned to Al-rich at ~440 °C, and the other assigned to a$_g$ZIF-62 at ~318 °C by comparing with that of the ZIF-62 control (Supplementary Fig. 3). The (ZIF-62)(Al-rich)(50/50) sample heated for 30 minutes at 450 °C showed almost identical behaviour; the first upscan showed an endotherm from ZIF-62 melting at 428 °C followed by the inorganic glass transition. As with the '1 minute' sample, the second upscan contained two glass transitions, assigned to the inorganic, again at ~440 °C and a$_g$ZIF-62 at ~319 °C (Fig. 2b).

The ZIF-62 melting endotherm was not evident in DSC experiments on the (ZIF-62)(Base)(50/50) and (ZIF-62)(Na-deficient)(50/50) samples due to the overlap of the glass transition

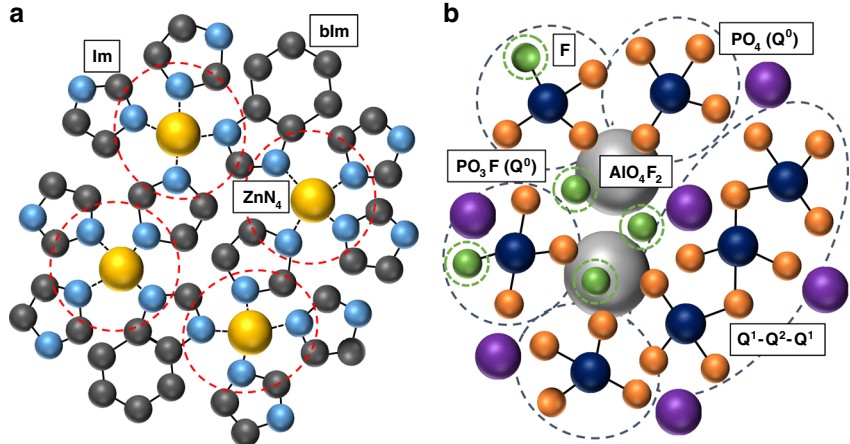

**Fig. 1 Structural chemistries.** 2D representations of **a** The structure of ZIF-62, showing $ZnN_4$ tetrahedra connected by imidazolate (Im) and benzimidazolate (bIm) organic ligands and **b** the local structure of the $(1 − x)([Na_2O]_z[P_2O_5])-x([AlO_{3/2}][AlF_3]_y)$ glass series, where $z$ and $y$ represent 1:$z$ and 1:$y$ ratios of $P_2O_5$: $Na_2O$ and $AlO_{3/2}$: $AlF_3$, respectively. The inorganic glasses are composed of varying lengths of phosphate tetrahedra chains and Al$(OP)_4F_2$ octahedra with $Na^+$ as the counter cation. Key: N—light blue, Zn—yellow, C—dark grey, P—dark blue, O—orange, Al—light grey, F—green, Na—purple, H—omitted for clarity. Inorganic structure adapted from Le et al.[22] with permission.

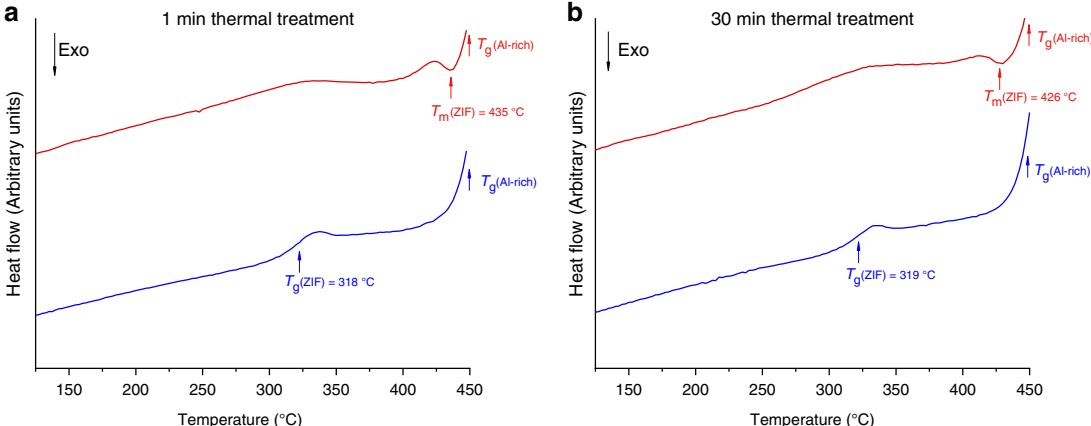

**Fig. 2 Glass transition behaviour.** Thermal response of (ZIF-62)(Al-rich)(50/50) on the first (red) and second (blue) DSC heating scans. **a** (ZIF-62)(Al-rich)(50/50)–1 min sample. **b** (ZIF-62)(Al-rich)(50/50)−30 min sample. An offset has been added to improve readability.

of the inorganic glass component with the melting point of ZIF-62. However, the second heating cycle of the (ZIF-62)(Na-deficient)(50/50) and (ZIF-62)(base)(50/50) samples did contain separate $a_g$ZIF-62 and inorganic glass transitions irrespective of the length of time spent at 450 °C (Supplementary Figs. 5 and 6). Thermogravimetric analysis (TGA) confirmed that none of the samples had any substantial mass loss upon heating to 450 °C (Supplementary Fig. 7).

Bulk samples of the composites were then prepared in consideration of these DSC results, by heating pressed pellets (see methods) of the three (ZIF-62)(Inorganic Glass)(50/50) powders in a tube furnace heated to 410 °C for 1 minute, and, in a separate experiment, for 30 minutes. This lower temperature is still greater than the onset of melting for ZIF-62, and was used due to the much slower cool, and therefore longer time that was spent at elevated temperatures for the tube furnace samples. The samples formed upon cooling in each case are referred to as $(a_g$ZIF-62$)_{0.5}$(Inorganic Glass)$_{0.5}$–1 min and $(a_g$ZIF-62$)_{0.5}$(Inorganic Glass)$_{0.5}$–30 min respectively.

**Surface characterisation.** Confocal microscopy was used to characterise the surface of the samples (Fig. 3, Supplementary

Fig. 8). Clear evidence of flow in all cases was observed, with heat treatment for longer periods of time resulting in grain growth, reduction of interfaces and increased light transmittance through the samples. In addition, features indicating the action of surface tension were found, such as rounding of grains to form 'islands' and spheroidal bubbles[28]. Large droplets lying on the surface, and, particularly, smooth surfaces over large areas are visible in the top-lit microscope images (Supplementary Fig. 8). Laser scanning microscopy was used to measure the roughness profile parameters (Supplementary Fig. 9), the arithmetical mean deviation of the primary profile decreases with heating time in all sample compositions, in accordance with the optical results. Given the sensitivity of nanoindentation to surface roughness, the surface profiles of $(a_g$ZIF-62$)_{0.5}$(Na-deficient)$_{0.5}$ and $(a_g$ZIF-62$)_{0.5}$(Al-rich)$_{0.5}$ (Supplementary Table 2) confirmed that they were unsuitable for the technique, however the $(a_g$ZIF-62$)_{0.5}$(base)$_{0.5}$ samples were examined using nanoindentation (Fig. 4). The $(a_g$ZIF-62$)_{0.5}$(base)$_{0.5}$ results show clear heterogeneity in the sample, even on the 100 μm scale, with regions of high, and low hardness ($H$) and modulus ($E$). Stiffness values for pure samples of $a_g$ZIF-62 and the base inorganic glass are ~6.6 GPa[29] and 51 GPa, respectively. The results show a significant decrease in heterogeneity in the samples heated for

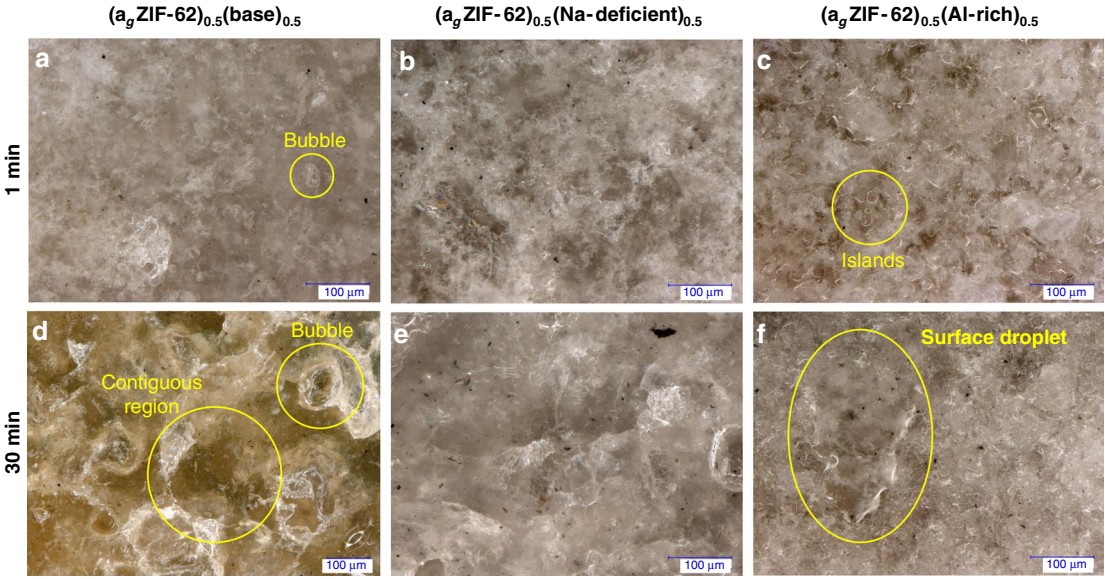

**Fig. 3 Glass flow.** Side-illuminated *z*-scan digital microscopy images of the (a$_g$ZIF-62)$_{0.5}$(Inorganic Glass)$_{0.5}$ compositions heat treated for 1 minute **a**–**c** and 30 minutes **d**–**f**.

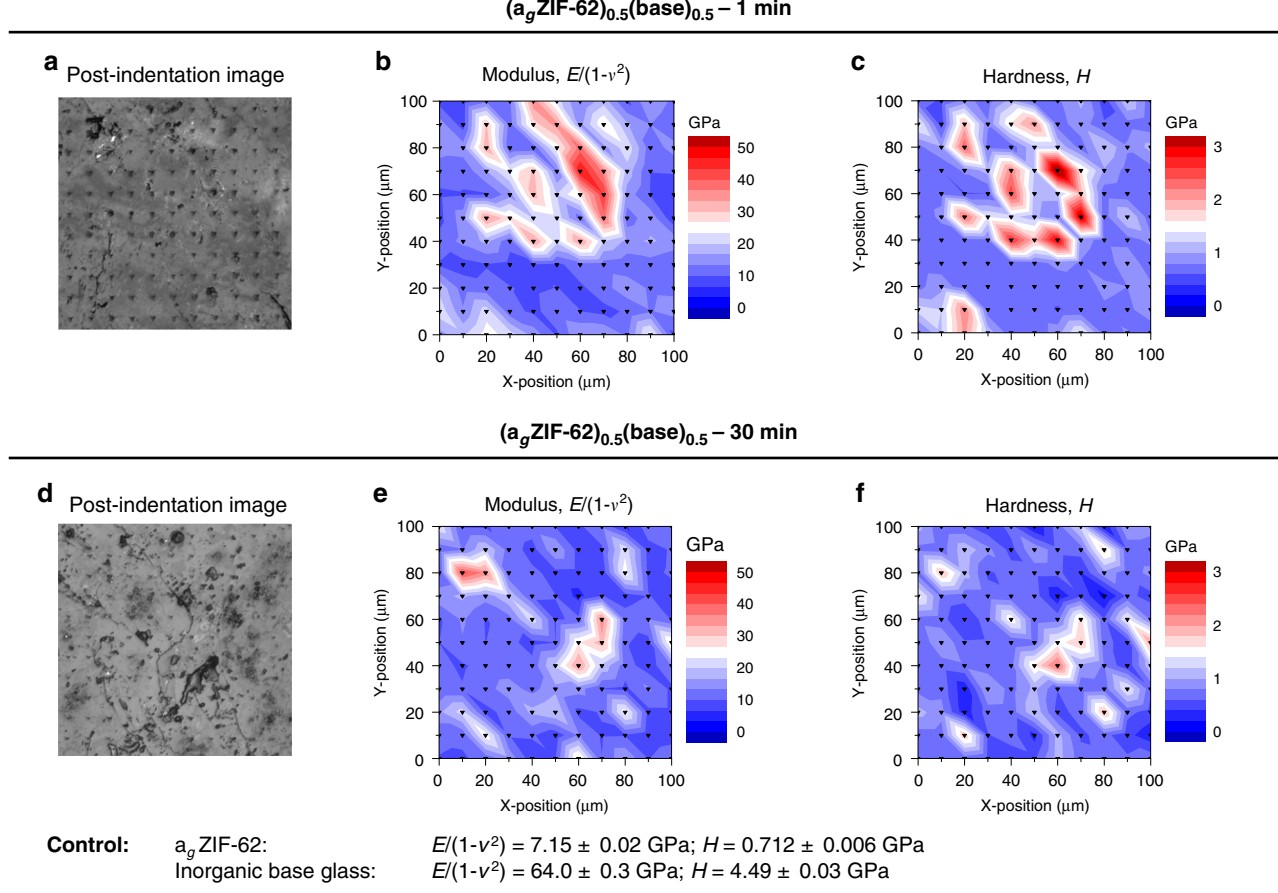

| Control: | a$_g$ZIF-62: | $E/(1-v^2) = 7.15 \pm 0.02$ GPa; $H = 0.712 \pm 0.006$ GPa |
|---|---|---|
|  | Inorganic base glass: | $E/(1-v^2) = 64.0 \pm 0.3$ GPa; $H = 4.49 \pm 0.03$ GPa |

**Fig. 4 Mechanical properties variation.** Modulus (*E*) and hardness (*H*) contour maps together with wide-field confocal microscopy images of the area mapped across the surface of the (a$_g$ZIF-62)$_{0.5}$(base)$_{0.5}$ compositions heat treated for 1 minute **a**–**c** and 30 minutes **d**–**f**.

30 minutes, in agreement with the surface profile parameters. This is accompanied by a decrease in the average *E*, suggesting a more compliant structure is formed upon mixing MOF and inorganic glass.

The scratch resistance of the (a$_g$ZIF-62)$_{0.5}$(base)$_{0.5}$–1 min sample, a$_g$ZIF-62, and the inorganic base glass was also investigated (Fig. 5). Owing to the lower hardness of a$_g$ZIF-62 ($H$(a$_g$ZIF-62) = 0.71 GPa) as compared with the inorganic glass

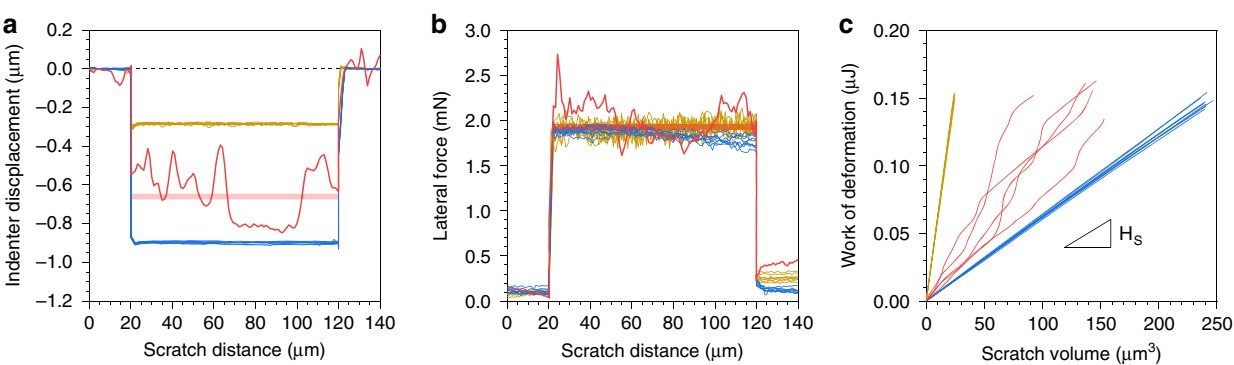

**Fig. 5 Scratch resistance. a** Spatial variation in indenter displacement ($h$). **b** Spatial variation in lateral force ($F_L$) as monitored during a constant-load scratch test across the surface of the samples. **c** Work of deformation ($W_S$), as derived from the integral of $F_L$ across 80 μm, neglecting the first and last 10 μm of each scratch, versus the corresponding scratch volume ($V_S$) of the investigated glass samples. The scratch hardness ($H_S$) is defined as the slope of the linear regression curve of $W_S$ versus $V_S$. Key: ($a_g$ZIF-62)$_{0.5}$(base)$_{0.5}$–1 min—red, base—gold and $a_g$ZIF-62—blue. Red shades display the confidence intervals (95 %) for a) $h$ and b) $F_L$ as derived from multiple such scratch tests performed on the ($a_g$ZIF-62)$_{0.5}$(base)$_{0.5}$–1 min glass sample.

($H$(base) = 4.49 GPa) a considerably larger sample volume of $a_g$ZIF-62 is deformed during scratching (Fig. 5a). Despite this mismatch in the indenter displacement ($h$), very similar values of lateral force ($F_L$) were recorded for these two glasses (Fig. 5b), indicating a substantially lower resistance of $a_g$ZIF-62 against the lateral movement of the indenter tip, i.e., a lower scratch hardness[30], as compared with the inorganic base glass. When probing the ($a_g$ZIF-62)$_{0.5}$(base)$_{0.5}$–1 min glass sample, pronounced fluctuations in both $h$ and $F_L$ are clearly visible during scratching. The length scale of these variations corresponds very well to the microstructural scale of the composite constituents, an effect also observed in the mechanical resistance, i.e., hardness and modulus, of the composite material as revealed by nanoindentation (Fig. 4). Mean values of $h$ (Fig. 5a) and $F_L$ (Fig. 5b) for the ($a_g$ZIF-62)$_{0.5}$(base)$_{0.5}$–1 min glass sample are in-between the $a_g$ZIF-62 and inorganic base glasses, which confirms our earlier conclusion that the composite materials are, on average, more compliant than the inorganic base glass but mechanically more stable than pure $a_g$ZIF-62. The scratch hardness ($H_S$) represents the work, $W_S$, which is required to generate (deform) a scratch groove of volume $V_S$ (Fig. 5c)[30], the value of $W_S/V_S$ for the ($a_g$ZIF-62)$_{0.5}$(base)$_{0.5}$–1 min glass sample is consistently above that of pure $a_g$ZIF-62 ($H_S$($a_g$ZIF-62) = 0.45 GPa), and close to that of the pure inorganic base glass ($H_S$(base) = 4.84 GPa).

**Spectroscopy.** [1]H nuclear magnetic resonance (NMR) spectroscopy showed no appreciable changes in the organic linker ratio upon heating the ($a_g$ZIF-62)$_{0.5}$(Inorganic Glass)$_{0.5}$ samples for 1 minute. However, in the ($a_g$ZIF-62)$_{0.5}$(Inorganic Glass)$_{0.5}$–30 min samples the [bIm]/[bIm+Im] ratio was 0.3% higher, implying a common equilibrium state independent of the inorganic composition (Supplementary Figs. 10–12, Supplementary Tables 3–9).

[31]P MAS NMR spectroscopy was also carried out to investigate changes in the phosphate component of the inorganic glass. Peaks in the ($a_g$ZIF-62)$_{0.5}$(Inorganic Glass)$_{0.5}$–1 min and –30 min samples were found at higher chemical shifts than their respective pure inorganic glasses (with the exception of the Na-deficient–1 min composition) (Supplementary Figs. 13–15). New intensity in the [31]P spectra appeared in the region 5 to −15 ppm and increased proportionally with heat treatment time (Fig. 6, Supplementary Figs. 16 and 17). This is consistent with literature values for the shifts of [31]P in PO$_3$N and PO$_2$N$_2$ species at −10 and 0 ppm, respectively[31], indicating the possibility of P–N bond

formation between the phosphate tetrahedra and Im ring. [31]P {[1]H} cross polarisation (CP) NMR measurements were performed of the same samples (Fig. 6, Supplementary Figs. 16 and 17) to further interpret this additional intensity. CP NMR experiments measure the proximity of nuclei in space; the efficiency of the transfer of magnetisation is mediated by the dipolar coupling of heteronuclear spins (which has an $r^{-3}$ dependence)[32]. Thus, [31]P {[1]H} CP NMR experiments (Fig. 6, Supplementary Figs. 16 and 17) shed light on these new peaks found in the 1d [31]P spectra (highlighted by the residuals), specifically they reveal that the new intensity in the 1d [31]P spectra can be assigned to phosphorus atoms with protons nearby. Despite no discernible intensity in the corresponding 1d [31]P spectra, there is a peak centred at ~12 ppm in the CP NMR spectra, which is in the ppm range of a phosphate tetrahedra without any bridging P–O–P bonds. The efficiency of magnetisation transfer from [1]H to [31]P (as evidenced by increased intensity in the CP spectra) is commensurate with the proximity and number of nearby protons, thus, this new peak at ~12 ppm may reflect that an isolated phosphate tetrahedron is more mobile within the composite, and therefore, is found close to the protons of the imidazolate or benzimidazolate rings. All of the samples exhibit an increase in the intensity of the peaks in the [31]P{[1]H} CP NMR spectra as a function of heat treatment time, including the peak located at approximately 12 ppm (Fig. 6, Supplementary Figs. 16 and 17).

Infra-red (IR) spectroscopy confirmed the integrity of the $a_g$ZIF-62 component within the composites (Supplementary Figs. 18–21). However, there were consistent small changes in the ~700 and 1450 cm$^{-1}$ peaks, which have been assigned to out-of-plane bending and ring stretching indicating some added deformation due to the presence of inorganic glass.

Raman spectra for $a_g$ZIF-62–1 min and –30 min samples were nearly identical (Supplementary Figs. 22 and 23), aside from a slightly larger redshift of the Zn–N peak at ~175 cm$^{-1}$ in the 30 min sample. Raman spectra for the ($a_g$ZIF-62)$_{0.5}$(Inorganic Glass)$_{0.5}$–1 min and –30 min samples contained similar features, ascribed to the $a_g$ZIF-62 component (Supplementary Fig. 24). The most significant change was in the low frequency Zn–N region (~175 cm$^{-1}$) where a second peak emerges at ~145 cm$^{-1}$ (Supplementary Fig. 25). We link the reaction to the formation of new Na–N bonds, given similar peaks in sodium imidazolate-containing compounds at 161 and 136 cm$^{-1}$ [33], and an as-purchased pure compound sodium imidazolide derivative (strong peak at 150 cm$^{-1}$) (Supplementary Fig. 26).

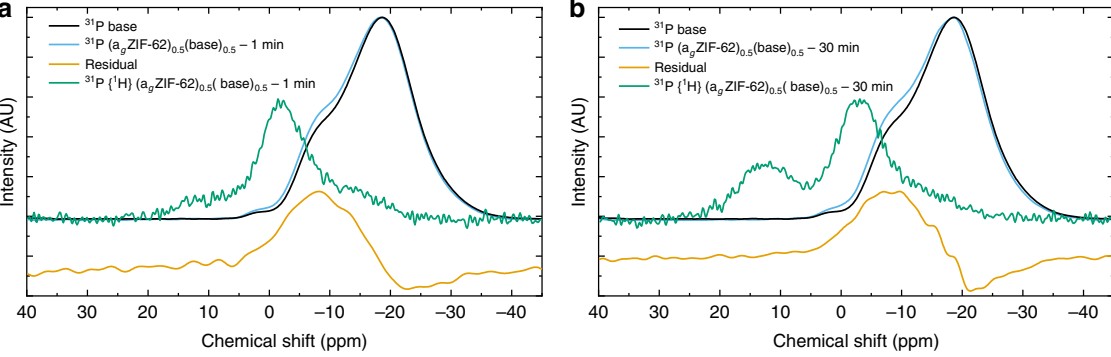

**Fig. 6 NMR spectroscopy.** $^{31}$P SS NMR and $^{31}$P{$^1$H} CP NMR of the inorganic glass (base) and the composites ($a_g$ZIF-62)$_{0.5}$(base)$_{0.5}$ samples. **a** ($a_g$ZIF-62)$_{0.5}$(base)$_{0.5}$–1 min and **b** ($a_g$ZIF-62)$_{0.5}$(base)$_{0.5}$−30 min.

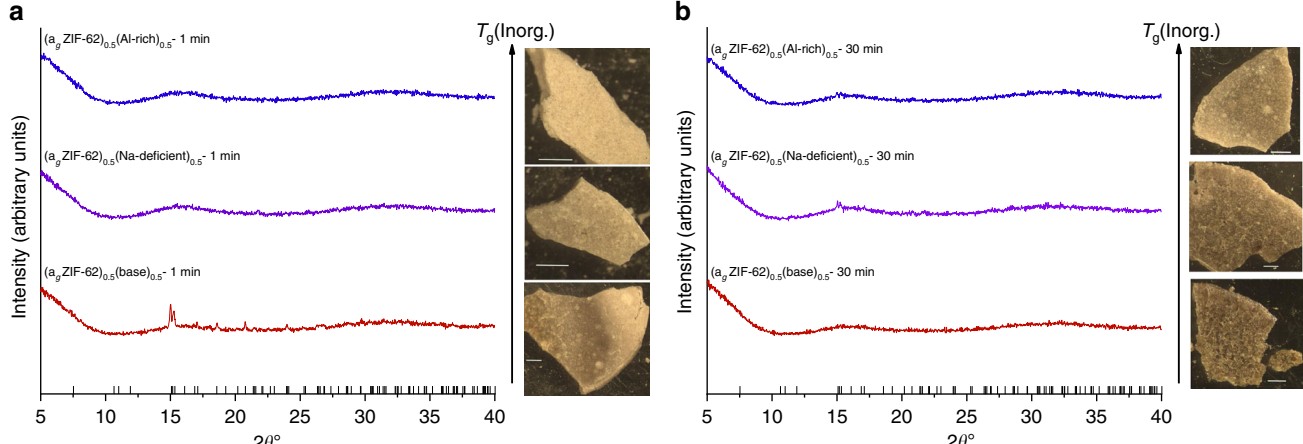

**Fig. 7 X-ray diffraction patterns and light microscopy.** PXRD and light microscopy (inset). **a** ($a_g$ZIF-62)$_{0.5}$(Inorganic Glass)$_{0.5}$–1 min. **b** ($a_g$ZIF-62)$_{0.5}$(Inorganic Glass)$_{0.5}$−30 min. Black check marks show positions of peaks in ZIF-zni[34]. An offset has been added to improve readability. The white bar in each image is 1 mm.

The ($a_g$ZIF-62)$_{0.5}$(Inorganic Glass)$_{0.5}$−30 min samples also converge to a consistent C–N peak position (~1175 cm$^{-1}$), red-shifted relative to the $a_g$ZIF-62−30 min control (Supplementary Fig. 27). No discernible features arising from the inorganic glass were able to be unambiguously determined, however.

EDS was also used to investigate microstructure. The locations of the ZIF-62 glass domains were identified using the zinc signal, whereas those from the inorganic glass were determined by signals from aluminium and phosphorus. In each case the heaviest elements from each component, Zn(Im)$_2$, NaPO$_3$ and AlF$_3$, were used to obtain the clearest signal. In all composite samples, distinct, segregated domains could be seen (Supplementary Figs. 28–33).

**X-ray diffraction and microscopy.** Powder X-ray diffraction (PXRD) on (ZIF-62)(Inorganic Glass)(50/50) samples confirmed that the crystal structure of ZIF-62 was intact prior to heat treatment (Supplementary Fig. 34). The ($a_g$ZIF-62)$_{0.5}$(Na-deficient)$_{0.5}$–1 min and ($a_g$ZIF-62)$_{0.5}$(Al-rich)$_{0.5}$–1 min PXRD patterns appeared completely amorphous. The PXRD pattern of the ($a_g$ZIF-62)$_{0.5}$(base)$_{0.5}$–1 min however contained a small number of low intensity Bragg peaks (Fig. 7a). The positions of these peaks, and in particular the most intense pair at ~15° 2θ, were found to match the reference pattern for ZIF-zni, a dense zinc imidazolate (Zn(Im)$_2$) framework, reported in the

literature[34]. The two Bragg reflections at ~15°, which are ascribed to the {400} and {112} reflections from ZIF-zni, were also present in the PXRD patterns of the ($a_g$ZIF-62)$_{0.5}$(Na-deficient)$_{0.5}$−30 min and ($a_g$ZIF-62)$_{0.5}$(Al-rich)$_{0.5}$−30 min samples. In contrast, the diffraction pattern of ($a_g$ZIF-62)$_{0.5}$(base)$_{0.5}$−30 min appeared to be completely amorphous (Fig. 7b).

The macroscopic appearance of the bulk composites was recorded using reflected light microscopy. ($a_g$ZIF-62)$_{0.5}$(Na-deficient)$_{0.5}$–1 min and ($a_g$ZIF-62)$_{0.5}$(Al-rich)$_{0.5}$–1 min were white, and appeared to be sintered powder bodies, though ($a_g$ZIF-62)$_{0.5}$(Na-deficient)$_{0.5}$–1 min was darker and more glassy in appearance. In contrast to this ($a_g$ZIF-62)$_{0.5}$(base)$_{0.5}$–1 min was inhomogeneous in appearance, with both white 'sintered powder' regions, darker 'macroscopically glassy' regions and distinct 'orange' regions (Fig. 7). The ($a_g$ZIF-62)$_{0.5}$(Na-deficient)$_{0.5}$−30 min and ($a_g$ZIF-62)$_{0.5}$(Al-rich)$_{0.5}$−30 min samples looked generally darker and more macroscopically glassy. However, ($a_g$ZIF-62)$_{0.5}$(base)$_{0.5}$ − 30 min appeared different in appearance from ($a_g$ZIF-62)$_{0.5}$(base)$_{0.5}$–1 min, with evidence of large pores and cracking in the pellet (Fig. 7b insert). Scanning electron microscopy experiments (SEM) performed confirmed these results, showing an increase in homogeneity on heat treatment time, particularly in the ($a_g$ZIF-62)$_{0.5}$(Al-rich)$_{0.5}$ samples. SEM also revealed that despite macroscopic cracking and pores, large regions of the ($a_g$ZIF-62)$_{0.5}$(base)$_{0.5}$−30 min remain smooth and homogeneous (Supplementary Figs. 35–37).

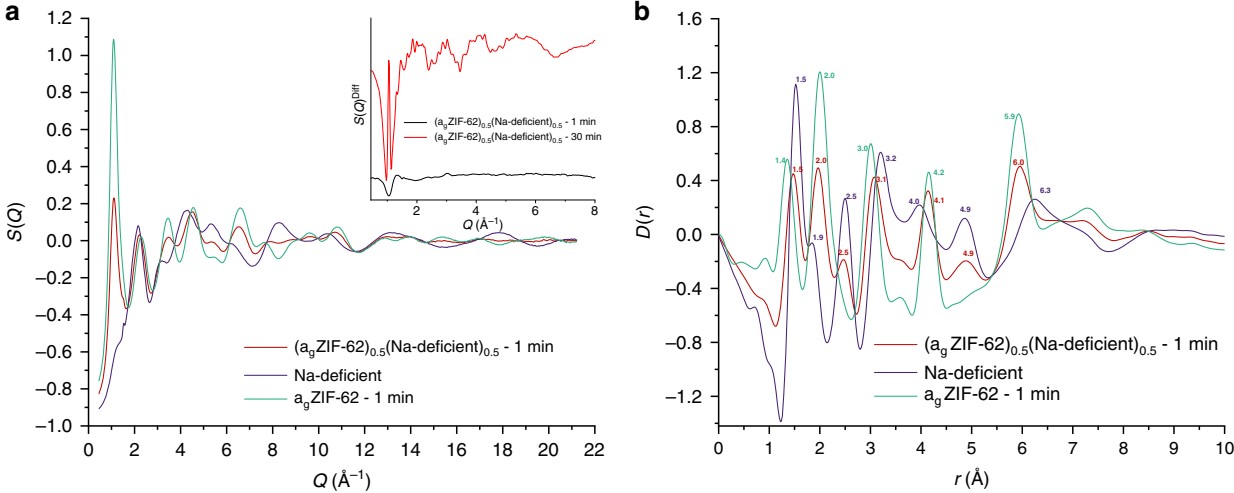

**Fig. 8 X-ray PDF of (a$_g$ZIF-62)$_{0.5}$(Na-deficient)$_{0.5}$–1 min. a** Total scattering of (a$_g$ZIF-62)$_{0.5}$(Na-deficient)$_{0.5}$–1 min, a$_g$ZIF-62–1 min and Na-deficient glass, insert: sharpened difference of (a$_g$ZIF-62)$_{0.5}$(Na-deficient)$_{0.5}$–1 min and (a$_g$ZIF-62)$_{0.5}$(Na-deficient)$_{0.5}$–30 min samples with an offset applied for clarity (see main text). **b** X-ray pair distribution function of (a$_g$ZIF-62)$_{0.5}$(Na-deficient)$_{0.5}$–1 min, a$_g$ZIF-62–1 min and Na-deficient glass.

**Pair distribution function studies**. To further investigate the structure of the composites, total scattering experiments were conducted on the (a$_g$ZIF-62)$_{0.5}$(Inorganic Glass)$_{0.5}$–1 min and (a$_g$ZIF-62)$_{0.5}$(Inorganic Glass)$_{0.5}$–30 min samples, as well as the pure inorganic glasses and a$_g$ZIF-62 controls (Fig. 8a, Supplementary Figs. 38–42). Consistent with the observations from the PXRD data (Fig. 7) the total scattering structure factors, $S(Q)$, of the (a$_g$ZIF-62)$_{0.5}$(base)$_{0.5}$–1 min, (a$_g$ZIF-62)$_{0.5}$(Al-rich)$_{0.5}$–30 min and (a$_g$ZIF-62)$_{0.5}$(Na-deficient)$_{0.5}$–30 min samples contained small Bragg peaks ascribed to ZIF-zni, in particular the {400} and {112} peaks (Supplementary Figs. 38–42). In addition to this, the Na-deficient glass and (a$_g$ZIF-62)$_{0.5}$(Na-deficient)$_{0.5}$ samples also contained a single weak Bragg peak, indicating a small amount of recrystallisation occurred on formation of the inorganic glass itself (Fig. 8a, Supplementary Fig. 40). The corresponding pair distribution functions for the composites, obtained by Fourier transform of the total scattering data, contain peaks that, in the main, correspond to those found in the a$_g$ZIF-62 and inorganic glasses (Fig. 8b, Supplementary Figs. 43–48).

In order to identify whether the composites contained new correlations, a differential method was used (See Supplementary Methods)[35,36]. In brief, the expected scattering intensity of a non-interacting mixture of a$_g$ZIF-62 and the relevant inorganic glass was calculated from the measured total scattering of each pure sample. This was then compared with the total scattering data collected on the composite samples, with the difference between them corresponding to scattering owing to interaction between the a$_g$ZIF-62 and inorganic glass in the composite. The process reveals Bragg like peaks in the difference for the (a$_g$ZIF-62)$_{0.5}$(base)$_{0.5}$–1 min, (a$_g$ZIF-62)$_{0.5}$(Na-deficient)$_{0.5}$–30 min and (a$_g$ZIF-62)$_{0.5}$(Al-rich)$_{0.5}$–30 min samples (Fig. 8a insert, Supplementary Figs. 49–54), in line with those seen in the laboratory diffraction data. Interestingly, this method also reveals weak remnant Bragg peaks in the (a$_g$ZIF-62)$_{0.5}$(base)$_{0.5}$–30 min sample (Supplementary Fig. 50), which were not observable in the PXRD data owing to the improved statistics of the synchrotron source. These features are obscured in the $S(Q)$ owing to the most intense Bragg diffraction peak coinciding with the first sharp diffraction peak from the a$_g$ZIF-62.

Real space data, $D(r)^{Diff}$, were obtained by Fourier transform of the structure factor, $S(Q)^{Diff}$, corresponding to these intensity differences (Supplementary Figs. 55–60). However, the $D(r)^{Diff}$ of

all the samples contain residual features due to the inorganic glass, and/or ZIF-62 or ZIF-zni $D(r)$s. Moreover, no correlations that could be definitively ascribed to new bonds observed through Raman scattering or $^{31}$P NMR data could be observed (Supplementary Figs. 57 and 59). These observations are explained by the unexpected change in the nature of the ZIF component upon heating (Fig. 8a insert, Supplementary Figs. 55 and 56), alongside the low interfacial interaction volume, meaning that new correlations may be below the detectable limit of the technique. The $D(r)^{Diff}$ of the (a$_g$ZIF-62)$_{0.5}$(Inorganic Glass)$_{0.5}$–30 min and the (a$_g$ZIF-62)$_{0.5}$(base)$_{0.5}$–1 min are, however, all qualitatively similar as expected from the similar Bragg scattering observed in the $S(Q)^{Diff}$ (Supplementary Figs. 57–60), which is attributed to the formation of ZIF-zni in the heat-treated composite samples.

Long-range order was also evident in the $D(r)^{Diff}$ from (a$_g$ZIF-62)$_{0.5}$(base)$_{0.5}$–1 min, (a$_g$ZIF-62)$_{0.5}$(Na-deficient)$_{0.5}$–30 min and (a$_g$ZIF-62)$_{0.5}$(Al-rich)$_{0.5}$–30 min samples. However, the $D(r)^{Diff}$ (a$_g$ZIF-62)$_{0.5}$(Base)$_{0.5}$–30 min sample appeared flat at extended distances, which is owing to the very small proportion of crystalline component as seen in the very small Bragg features in the $S(Q)^{Diff}$ (Supplementary Figs. 58 and 60).

**Ionic conductivity measurements**. The indication from Raman spectroscopy, that Na$^+$ ions enter the a$_g$ZIF-62 domains of the (a$_g$ZIF-62)$_{0.5}$(Inorganic Glass)$_{0.5}$ samples, led us to perform ionic conductivity measurements on the (a$_g$ZIF-62)$_{0.5}$(Na-deficient)$_{0.5}$–1 min and 30 min samples. These were carried out in order to demonstrate the nature of the bonding and mobility of sodium ions in the composites, and how both were affected by heat treatment time.

The ionic conductivity was measured between 110–200 °C for the (a$_g$ZIF-62)$_{0.5}$(Na-deficient)$_{0.5}$ samples heat treated for 1 minute and 30 minutes, and between 50 and 250 °C for the pure Na-deficient glass (Supplementary Figs. 61–64) and the activation energy ($E_a$) for ion motion was extracted (Fig. 9). The a$_g$ZIF-62–1 min had measurements of $<10^{-10}$ S/cm, meaning that an accurate measurement of the conductivity and activation energy could not be obtained owing to the lack of mobile ions in the a$_g$ZIF-62 phase. Therefore, the a$_g$ZIF-62 conductivity is reported at the limit of $10^{-10}$ S/cm, representing an upper bound of conductivity (Fig. 9). Although the $E_a$ for sodium conduction in a$_g$ZIF-62 is

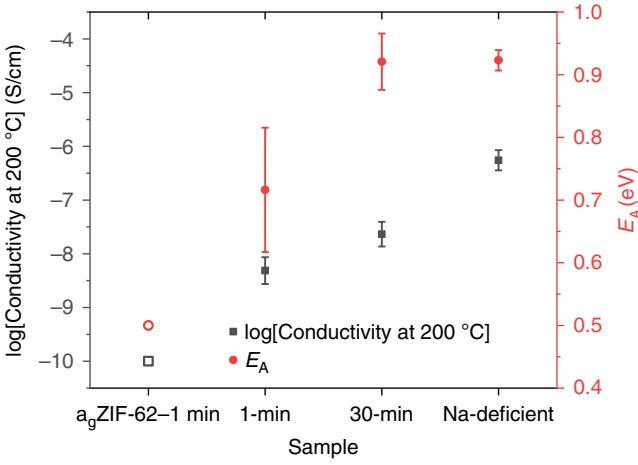

**Fig. 9 Ionic conductivity measurements.** Measurements of ionic conductivity at 200 °C of the $(a_gZIF-62)_{0.5}(Na\text{-deficient})_{0.5}-1$ min, $(a_gZIF-62)_{0.5}(Na\text{-deficient})_{0.5}-30$ min and Na-deficient samples along with the activation energy extracted from the gradient of the conductivity-temperature measurements (see methods). Data for $a_gZIF-62-1$ min are predictions only (see text) and are represented with open symbols.

unknown, as no $Na^+$ is present, the conductivity of ionic liquid impregnated amorphous ZIF-8 showed an $E_a$ of ~0.3 eV for $Na^+$[37], and therefore $a_gZIF-62$ is assumed to have a similar value.

The $(a_gZIF-62)_{0.5}(Na\text{-deficient})_{0.5}-1$ min sample showed a reduction in conductivity at 200 °C relative to the Na-deficient glass. This is explained by the addition of the non-conductive $a_gZIF-62$ phase, which reduces the concentration of sodium ions per volume. This is corroborated by the decrease in densities in the composite (Supplementary Tables 10–12). Furthermore, the microstructure of the $(a_gZIF-62)_{0.5}(Na\text{-deficient})_{0.5}-1$ min sample consists of remnant particles (Fig. 3, Supplementary Figs. 8 and 36), whose interfaces will act as defects reducing conduction. The conductivity of the $(a_gZIF-62)_{0.5}(Na\text{-deficient})_{0.5}-30$ min sample is larger, indicating that conductivity increases with annealing time. This is explained by densification (Supplementary Table 12) and the grain growth as evident from confocal microscopy and SEM (Fig. 3, Supplementary Figs. 8 and 36), indicating an enhanced $[Na^+]/cm^3$ as well as a more efficient sintering, with the latter resulting in a reduction in interfaces and defects.

The activation energy for ion motion is lower in the $(a_gZIF-62)_{0.5}(Na\text{-deficient})_{0.5}-1$ min sample than in the Na-deficient sample (i.e., the inorganic glass alone), indicating a low-energy pathway for $Na^+$ motion through the structure. As the Raman data indicate formation of Na–N bonding, the lower $E_a$ could indicate a potential motion of $Na^+$ ions through the $a_gZIF-62$ glass phase, with a lower activation energy owing to the phases more porous nature. The experimental error of this measurement may also be indicative of the large degree of structural heterogeneity observed in this sample by microscopy. However, after extended annealing time, the $(a_gZIF-62)_{0.5}(Na\text{-deficient})_{0.5}-30$ min sample shows an activation energy more like that of the bulk Na-deficient glass sample and with a reduced experimental error. The increase in $E_a$ may be due to the overall structural densification in which a higher energy but more prevalent conduction pathway through the Na-deficient glass phase predominates over interfacial conduction through $a_gZIF-62$ boundaries. Taken together, these results indicate that the composite samples show an appreciable degree of conduction of $Na^+$ ions, whose exact sodium conduction mechanisms are of interest to the active sodium-ion battery community.

## Discussion

[1]H liquid NMR, IR and TGA results confirm the integrity of the imidazolate and benzimidazolate linkers in all these composite materials. EDS results demonstrate no substantial overlap between separate domains of predominately zinc signal, originating from the ZIF-62, and areas with signal from both aluminium and phosphorous, which originate from the inorganic glass phase. This agrees with the observation of two glass transitions in the DSC. The presence of inorganic glass and $a_gZIF-62$ domains measured in EDS are also in good agreement with the variations in $E$ and $H$ measured by nanoindentation mapping. These results indicate a structure of separate $a_gZIF-62$ and inorganic domains, which electron microscopy confirms are bonded at their interfaces into a single body.

The extent of interfacial mixing between the two phases is highly dependent upon the glass transition of the inorganic component. SEM of the $(a_gZIF-62)_{0.5}(base)_{0.5}$ ($T_g(base) = 372$ °C) samples showed a more homogeneous appearance than for those samples containing inorganic glasses with higher $T_g$. Indeed, the low degree of flow meant remnant particles were visible for $(a_gZIF-62)_{0.5}(Na\text{-deficient})_{0.5}-1$ min ($T_g(Na\text{-deficient}) = 414$ °C) and $(a_gZIF-62)_{0.5}(Al\text{-rich})_{0.5}-1$ min ($T_g(Al\text{-rich}) = 449$ °C). These results are intuitive, since the low temperature end of the glass transition can be described empirically as when a fluid has the viscosity of a solid ($\sim10^{15}$ Pa·s)[38], lower-$T_g$ inorganic glasses will have a lower viscosity at the same heat treatment temperature and therefore encourage a greater extent of intermixing between domains of different chemical compositions.

The X-ray diffraction and SEM experiments performed on a sample of $(a_gZIF-62)_{0.5}(base)_{0.5}-1$ min indicate a small degree of recrystallisation to the dense $[Zn(Im)_2)]$ polymorph, ZIF-zni (Supplementary Fig. 65)[34]. Continued isothermal treatment results in subsequent reduction of the ZIF-zni phase in the $(a_gZIF-62)_{0.5}(base)_{0.5}-30$ min sample. This reduction in Bragg scattering was further confirmed by PDF with the $S(Q)^{Diff}$, confirming that sharp Bragg features were still present in $(a_gZIF-62)_{0.5}(base)_{0.5}-30$ min, though to a smaller degree. The $(a_gZIF-62)_{0.5}(Al\text{-rich})_{0.5}-30$ min and $(a_gZIF-62)_{0.5}(Na\text{-deficient})_{0.5}-30$ min samples contained Bragg peaks ascribed to ZIF-zni (Supplementary Fig. 65), though in contrast, the $(a_gZIF-62)_{0.5}(Al\text{-rich})_{0.5}-1$ min and $(a_gZIF-62)_{0.5}(Na\text{-deficient})_{0.5}-1$ min did not.

ZIF-zni is reported to recrystallise from ZIF-4, a $Zn(Im)_2$ polymorph sharing the same cag topology as ZIF-62[39,40], on heating to ~370 °C before melting at ~590 °C[21,41]. The absence of recrystallisation in ZIF-62 has been ascribed to the bulkier benzimidazolate linker imposing added steric constraints on the $ZnN_4$ coordination polyhedra[21]. We therefore postulate that ZIF-zni formation arises in this case owing to an interaction between the inorganic and MOF glass phases, with stronger interactions occurring at lower viscosities of the inorganic glass component. This may proceed via migration of benzimidazole to the inorganic glass, which is consistent with prior literature showing that benzimidazole and zinc metaphosphate glass are miscible[42]; recrystallisation to ZIF-zni of the remnant Im-rich interface domains then occurs, before this itself either melts, or is dissolved by the melt on further heating in the base 30 minute sample. The effect is most pronounced in those samples with lower glass transition temperatures and hence lower viscosities at the treatment temperatures, which promote a greater degree of mixing.

The emergence of a large new peak at ~145 $cm^{-1}$ in the Raman spectra of all the $(a_gZIF-62)_{0.5}(Inorganic Glass)_{0.5}$ samples, which was ascribed to the formation of Na–N bonds, provides useful information on the interaction between the two phases. [31]P MAS NMR spectroscopy recorded for the $(a_gZIF-62)_{0.5}(Inorganic Glass)_{0.5}-1$ min and $-30$ min samples has a noticeable peak shift to higher ppm when compared with their respective inorganic

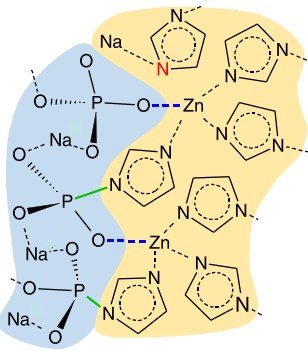

**Fig. 10 Potential interface structure.** Schematic of the interface between Inorganic glass and MOF glass domains, based on PDF, Raman and NMR spectroscopy.

glasses (with the exception of (a$_g$ZIF-62)$_{0.5}$(Na-deficient)$_{0.5}$−1 min, which had an initial increase in the lower ppm region, ~−26 ppm). In the literature, such shifts of $^{31}$P NMR peaks to higher ppm have generally been attributed to formation of terminal oxygens, causing a decrease in the average charge density on the phosphorus atoms[43]. However, here the chemistry of the system and preparation method means that we do not expect the creation of new terminal oxygen bonds at high ppm. An alternative explanation is the formation of P–N bonds; we would expect P–N bonds to markedly shift the average $^{31}$P peak positions to higher chemical shift. Furthermore, in a $^{31}$P study of phosphorus oxynitride glasses, it was found that PO$_3$N and PO$_2$N$_2$ peaks appear at −10 and 0 ppm, respectively[31]. Second, the $^{31}$P{$^1$H} CP spectra detect protons in the proximity of these phosphorus atoms located in the high ppm region. Consequently, the formation of additional peaks at high ppm in the $^{31}$P NMR spectra and good agreement with the $^{31}$P{$^1$H} CP spectra points toward a significant interaction between the Im$^-$ and bIm$^-$ linkers and phosphorus in the inorganic glass via P–N bond formation.

We therefore tentatively propose the schematic (Fig. 10) as one possible structure for the interface between the inorganic and ZIF glasses, in the composites formed here. The melting process of pure-phase ZIFs has previously been shown to involve Zn–N bond breakage at a critical temperature, which leaves both under-coordinated Zn, and relatively electron-rich N sites[14]. Sodium is known to be relatively mobile in inorganic glasses, especially at temperatures near $T_g$, and would be expected to migrate to atoms with extra electron bond density. The Raman data here indicate that N–Na coordination happens very early, with the NMR data being consistent with the establishment of an equilibrium state involving P–N bond formation and/or creation of terminal oxygen. Zn-O-P correlations, though not directly experimentally measured, were included for reasons of charge balance and to maintain tetrahedral coordination of Zn centres, their inclusion is also justified by the large number of examples of inorganic glasses, which contain similar structures[43–45]. We note that unfortunately, the Na Kα edge (1.040 keV) and Zn Lα edges (1.012 keV) are too close in energy to observe simultaneously using EDS[46], so we are unable validate the Raman results with elemental mapping.

These results describe a new class of inorganic–MOF glass composites, prepared by heating a mixture of a phosphate glass and ZIF-62. The composites formed upon cooling contain two distinct glass transition temperatures, matching those of a$_g$ZIF-62 and the relevant inorganic glass, implying that the composite contains separate domains of each glass phase bonded at their interface into a single solid body in agreement with SEM, mechanical and conductivity results. The extent of mixing is

dominated by the inorganic glass transition temperature, which is itself linked to the chemistry of the glass. The extent of mixing is great enough that it enables a reaction of the inorganic and MOF components to occur. This results in a small degree of recrystallisation of ZIF-62 to form a dense ZIF-zni phase; the precise nature of this interaction was not determined due to its limited extent but would be an interesting subject for further study.

The formation of materials containing interlocked inorganic glass and MOF glass will prove of great interest as prototypical examples of a new materials family, with mechanical and electrical properties intermediate between the two parent structures. The emergence of this new class of composites implies the ability to alter the physical, chemical and electrical properties of the vast array of inorganic glasses currently used in, e.g., display technologies and protective coatings. Critically the authors hope that the characterisation work here indicates an approach by which other researchers may explore this new class of composite materials.

## Methods

**Synthesis**. *ZIF-62*: crystalline ZIF-62 was synthesised according to the following method: zinc nitrate hexahydrate (1.65 g, 5.55 mmol), imidazole (8.91 g, 131 mmol) and benzimidazole (1.55 g, 13.12 mmol) were added to *N,N*-dimethylformamide (DMF) (75 ml). The mixture was then heated at 130 °C and stirred for 48 hours. The resultant product was washed with DMF (2 × 20 ml) under vacuum to obtain a white crystalline powder (yield 42.6%). To increase the yield the filtered reaction mixture was placed back into the oven at 130 °C for a further 48 hours and then more product obtained through washing under vacuum. For the heat-treated samples a mixture of the two filtrations was used to obtain enough ZIF-62. The ZIF-62 used in the controls and evacuated powder mixtures were synthesised in a separate synthesis where only the powder from the first filtration was used. PXRD (Supplementary Fig. 2) and NMR (Supplementary Figs. 10–12 and Supplementary Tables 3–9) confirmed that the products of both syntheses were the same.

Before direct experiments on the ball-milled powder mixtures could be conducted, the powders were activated by soaking in dichloromethane for 24 hours, followed by heating to 175 °C for 3 hours under vacuum. This was done to remove framework-templating DMF from within the pores of the ZIF-62.

*Inorganic glass samples*: high purity reagents (optical grade) of NaPO$_3$ and AlF$_3$ were melted in a Pt crucible in an electric muffle furnace. Owing to the known volatility of fluoride, care was taken to initially melt all mixtures at 800 °C for one hour to allow NaPO$_3$ to melt and dissolve the AlF$_3$ before higher temperatures were used for complete dissolution. Generally, longer melting times were preferred over higher melting temperatures when producing a homogeneous melt.

The base glass sample was melted at 800 °C for 1 hour before being taken up to 850 °C for half an hour before pouring. Higher amounts of AlF$_3$ required higher melting temperatures, with the Al-rich and Na-deficient glasses requiring 950 and 1000 °C to be completely homogeneous, respectively. Since the glasses were then to be pulverised and remelted, no attempts at annealing were conducted on the powders used for synthesis of the composites. Instead they were pulverised in a Retsch PM 100 grinder at 350 rpm with 1 min intervals for half an hour using ZrO$_2$ or Si$_3$N$_4$ balls (with roughly equal sample and ball volume). A bulk piece was saved from each composition to later be annealed for control measurements, such as elemental analysis and mechanical measurements. The annealing temperatures were 40–60 °C above the $T_g$ of the inorganic phase; the glass specimens were then cut and polished to one micron.

To make 80 g of the base inorganic glass, 66.3 g of dry NaPO$_3$ powder and 13.7 g of AlF$_3$ powder were mixed thoroughly by hand before melting. The Na-deficient composition was made from 59.1 g of dry NaPO$_3$ powder and 20.9 g AlF$_3$, whereas the Al-rich used 51.6 g and 28.4 g, respectively.

*Composite Samples [(a$_g$ZIF-62)$_{0.5}$(Inorganic)$_{0.5}$]*: approximately 300 mg of crystalline ZIF-62 and 300 mg of inorganic glass powders were mixed together through ball-milling in a stainless steel jar (15 ml) for 5 minutes at 25 Hz with one 5 mm stainless steel ball bearing in a Retsch MM400 grinder mill. In all composite samples, 200 mg samples of the ball-milled powder mixture were placed in a 13 mm die and compacted into a pellet using 10 tons of pressure applied for 1 minute. These pellets were placed in a tube furnace (Carbolite 12/65/550), which was left to equilibrate under argon for one hour before heating to 410 °C at 10 °C/min and holding for either 1 or 30 minutes. All heating was done under constant argon flow. The heat-treated pellets were left to cool under argon at the natural rate of the tube furnace; the samples were removed from the tube furnace at temperatures equal to or below 200 °C.

**Density**. *Archimedean method*: the densities of the inorganic glasses were measured 3–4 times by the Archimedes principle at RT in absolute ethanol.

*Pycnometry*: the densities of the crystalline and $a_g$ZIF-62, as well as the composites were measured using a Quantachrome Ultrapyc 1200e He pycnometer at 20.0 °C for 5 sets of 30 cycles each.

**Thermal characterisation.** *DSC*: DSC characterisation was conducted using a Netzsch 214 Polyma. Approximately 10 mg of sample was placed in aluminium crucibles with a pierced concave lid. Heating and cooling steps were conducted under argon at a rate of 10 °C/min. Features in the DSC traces were processed by smoothing and analysed using the Netzsch analysis software, with glass transition temperatures ($T_g$) calculated using the midpoint.

*TGA*: TGA curves were recorded using a TA instruments Q-600 series DSC. Approximately 10 mg powdered sample was placed in open alumina crucibles and heated at 10 °C/min under argon. The TGA data was analysed using the TA Universal Analysis software.

**Impedance spectroscopy.** The surface areas and thicknesses of the Na-deficient, $(a_g$ZIF-62$)_{0.5}$(Na-deficient)$_{0.5}$–1 min and $(a_g$ZIF-62$)_{0.5}$(Na-deficient)$_{0.5}$–30 min samples were measured (1 cm$^2$ and ~1 mm; ~2 mm$^2$ and 0.7 mm). All samples were well-polished; the Na-deficient sample was sputtered with a gold layer on both sides, however, the composites were left bare for electrical measurements.

The impedance measurements were performed on a Novocontrol Alpha-A spectrometer paired with a Novotherm Temperature Control System. The measured frequency range was from $10^{-1}$ to $10^7$ Hz. The temperatures from 50 to 250 °C with intervals of 25 °C were measured for the Na-deficient sample and for the ZIF samples $(a_g$ZIF-62$)_{0.5}$(Na-deficient)$_{0.5}$–1 min and $(a_g$ZIF-62$)_{0.5}$(Na-deficient)$_{0.5}$–30 min from 50 to 200 °C with intervals of 30 °C.

The resistance under direct current ($R_{DC}$) was determined as the right intersection of the $x$ axis with the half circle of the Nyquist Plot (real and imaginary part of the impedance, $Z'$ VS $Z''$), see Supplementary Figs. 61–63. The conductivity ($\sigma$) is calculated as:

$$\sigma = \frac{1}{R_{DC}} \frac{l}{A} \tag{1}$$

where $l$ is the thickness and $A$ is the area of the sample.

The temperature dependency of the ionic conductivity was described by the Arrhenius relation (Supplementary Fig. 64):

$$\sigma T = \sigma_0 \exp\left(-\frac{E_a}{K_B T}\right) \tag{2}$$

where $\sigma_0$ is the pre-factor, $K_B$ is the Boltzmann constant and $E_a$ is the activation energy of the ionic conductivity.

**Surface characterisation.** *Reflected light microscopy*: a Leica MZ95 microscope equipped with a Moticam camera with a resolution of 2 Mpixels was used to take reflected light microscopy images of the composite materials.

*Digital optical microscopy*: a Keyence VHX-6000 digital microscope equipped with VHX-H2MK software and VHX-500 3D Viewer 1.02 was used to optically image the samples. The camera is a CCD detector with a resolution of 54 Mpixels. Images were generated by focal scanning along the z-axis and image stacking. Photos with different lighting (top-lit vs. side-lit) and magnifications (×300, ×600 and ×1000) were taken.

*Confocal laser scanning microscopy*: the roughness of the samples was measured using a Carl Zeiss Imager-Z1m LSM 700 confocal scanning microscope (CLSM) with an Ar$^+$ laser (488 nm) and an 11 μm pinhole. CLSM increments the sample stage in the z-direction and stitches together the series of 'imaged' layers, where surface height differences create bright (focused) and dark (unfocused) regions. An ×20/0.50 HD objective was used to scan a square area of 320 μm$^2$, whereas the z axis range depended on the roughness of the surface. Roughness parameters were determined using the ZEN-black 2012 software. The error of the instrument is determined by the wavelength of the laser and pinhole size, which is expected to be on the order of 50 nm. Owing to surface inconsistencies in our samples, i.e., large isolated, random divets, we chose a representative line profile and determined the 2d roughness for this line rather than a global 3d surface roughness.

*Nanoindentation*: modulus ($E$) and hardness ($H$) mapping was performed at room temperature using a KLA Nanoindenter G200 equipped with a three-sided Berkovich diamond indenter tip. The tip area function and instruments frame compliance were calibrated prior to the first experiments on a fused silica reference glass specimen following the Oliver and Pharr method[47]. Indentations with a depth limit of 500 nm were performed at a strain rate of 0.05 s$^{-1}$. In total, 121 indents were created across an area of $100 \times 100$ μm$^2$ with a spacing of 10 μm between individual indentation marks. The values of $H$ were calculated from the load divided by the project contact area of the indenter tip at the maximum load and the values of $E$ were derived from the reduced modulus:

$$E_s = \left(1 - v_s^2\right)\left[\frac{1}{E_r} - \frac{\left(1 - v_i^2\right)}{E_i}\right]^{-1} \tag{3}$$

where $E$ and $v$ are the Young's modulus and Poisson's ratio, respectively, of the indenter tip (subscript '$i$') and the material tested (subscript '$s$'). Since the exact

values of $v_s$ of the individual phases present in the composite materials are unknown, we defined the modulus as:

$$E = \frac{E_s}{\left(1 - v_s^2\right)} \tag{4}$$

Optical micrographs of the indented surface area were recorded using a Zeiss Smartproof 5 wide-field confocal microscope.

*Scratch testing*: the scratch resistance was analysed in constant-load scratch tests using a three-sided Berkovich diamond tip in edge-forward orientation using the nanoindentation setup mentioned above. The indenter tip was moved across the sample surface along a distance of 100 μm at a fixed scratch velocity of 10 μm/s and under a prescribed normal load of 10 mN, while monitoring the lateral force ($F_L$) and indenter displacement ($h$). In total 10 such scratch tests were performed on each sample.

**NMR spectroscopy.** $^1H$ *NMR*: approximately 6 mg of powder was digested in a mixture of DCl (20%)/D$_2$O (0.1 ml) and DMSO-d$_6$ (0.6 ml) and the spectra recorded using a Bruker 500 MHz DCH Cryoprobe Spectrometer. Processing and analysis were conducted in TopSpin.

$^{31}P$ *MAS NMR*: $^{31}$P MAS NMR spectra were measured on a Bruker Avance III 400 (9.4 T magnet, 162 MHz for $^{31}$P) with a 4 mm MAS probe spinning at 12.5 KHz. All spectra were referenced to a non-spinning rotor filled with 85 wt% H$_3$PO$_4$. Quantitative single-pulse experiments were conducted with a 60° pulse length (2–2.5 μs) and delay times between 150 and 400 s. In cases when insufficient sample was available, Teflon tape was used to ensure the rotor was full before spinning.

**Total scattering measurements.** X-ray data were collected at the I15-1 beamline at the Diamond Light Source, UK ($\lambda = 0.161669$ Å, 76.7 keV). Samples were loaded into borosilicate capillaries of 1.17 mm inner diameter. Data on the samples, empty instrument and capillary were collected in the region of ~0.4 < $Q$ < ~22 Å$^{-1}$. Corrections for background, multiple scattering, container scattering, Compton scattering, fluorescence and absorption were performed using the GudrunX programme[48,49]. For further details on the differential method, please see methods in the SI.

**X-ray powder diffraction.** Data were collected using a B3 (BB) Bruker D8 DAVINCI diffractometer using Cu Kα ($\lambda = 1.5418$ Å) radiation and a LynxEye position sensitive detector in Bragg–Brentano parafocussing geometry. A 5-40° 2θ angular range was used with a step size of 0.02° and a step time of 0.75 s.

**Scanning electron microscopy and EDS.** Scanning electron microscopy and EDS were conducted on the composite samples using a FEI Nova NanoSEM. Samples were mounted on steel stubs using carbon tape and sputter coated with gold using a current of 20 mA for 2 min. Spectra were analysed using the Esprit software created by Bruker.

For the inorganic glasses, EDS was performed using a desktop SEM Phenom ProX instrument at 10 kV. The samples were fixed with an adhesive carbon tape on an aluminium sample holder.

**IR spectroscopy.** FTIR spectra of the powdered samples, ~5 mg, were collected on a Thermo Scientific Nicolet iS10 model FTIR spectrometer with an attenuated total reflection mode. All scans had a resolution of 2 cm$^{-1}$. A background scan was collected between each sample; almost no changes in the background were observed during collection.

**Raman spectroscopy.** The samples were embedded in epoxy and polished to 1 micron. Spectra were collected on Renishaw inVia Raman microscope at ×100 magnification using an excitation wavelength of 784 nm in a 180° scattering geometry; the resolution was 2 cm$^{-1}$ and the wavenumber region was 100–1500 cm$^{-1}$. The pure inorganic glasses were collected with higher laser power (100%) and long collection times (30 s) with a total of six scans. Similarly, the pure glassy ZIF controls were collected at 100% laser power, but due to fluorescence only 1 s collection time (180 scans) could be used without detector saturation; in the case of the 30 min heat treatment 30 s of bleaching was also required to prevent saturation. Longer bleaching times and more scans did not result in a better S/N ratio. The composite samples were significantly more fluorescent, therefore, requiring lower laser powers (5–10%) and longer bleaching times (up to 300 s was found to increase the S/N ratio). In general, lower $T_g$(inorg) composites needed the lowest laser powers and longest bleaching times, indicating the highest fluorescence. The technique was also found to be extremely sensitive to the surface quality with rough surface absorbing strongly. The resulting spectra were processed in the Renishaw software WiRe 4.0.

**Data availability**
The experimental data that support the findings of this study are available in Symplectic Elements with the identifier(s) https://doi.org/10.17863/CAM.58312.

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

## Acknowledgements

This project received funding from the European Research Council (ERC) under the European Union's Horizon 2020 research and innovation programme (ERC grant UTOPES, grant agreement no. 681652). T.D.B. acknowledges the Royal Society for a University Research Fellowship (UF150021), and a research grant (RSG\R1\180395). He wishes to thank the University of Canterbury Te Whare Wānanga o Waitaha, New Zealand, for a University of Cambridge Visiting Canterbury Fellowship, and the Leverhulme Trust for a Philip Leverhulme Prize (2019). Y.X. was funded by the Federal State of Thuringia and the EuropeanSocial Fund (ESF) through project HyNIB (2017FGR0055). L.L. acknowledges the EPSRC for a PhD studentship. J.M.T. acknowledges NanoDTC ESPSRC Grant EP/L015978/1. M.F.T. would like to thank Corning Incorporated for PhD. funding. A.F.S. acknowledges the EPSRC for a PhD studentship award under the industrial CASE scheme, along with Johnson Matthey PLC (JM11106). We acknowledge the provision of synchrotron access to Beamline I15-1 (EE20038), at the Diamond Light Source, Rutherford Appleton Laboratory, UK.

## Author contributions

L.L. and C.C. wrote the manuscript with the help of T.D.B. and L.W. Sample preparation and synthesis was done by L.L. and C.C. X-ray total scattering measurements were carried out by L.L. with the assistance of J.M.T., A.F.S., M.F.T. and D.S.K. Interpretation of the PDF results and development of the differential PDF measurements was carried out by D.A.K. and L.L. Nanoindentation and scratch testing experiments were carried out by R.L. DSC and TGA were carried out by L.L. X-ray diffraction was carried out by L.L. Reflected light microscopy, energy-dispersive spectroscopy and scanning electron microscopy was carried out by L.L. C.C. and Y.X. performed ion conductivity experiments. Raman and IR spectroscopy was carried out by C.C. Confocal microscopy was carried out by C.C. and L.W. NMR experiments and interpretation of the results were carried out by C.C.

## Competing interests

The authors declare no competing interests.

**Additional information**

**Peer review information** *Nature Communications* thanks the anonymous reviewers for their contributions to peer review of this work. Peer review reports are available.

