## [Peer Review File · Nature Communications]

REVIEWER COMMENTS

Reviewer #1 (Remarks to the Author):

This manuscript describes the synthesis, characterization and properties of mixed metal-organic framework and inorganic glass composites. Particularly, the authors chose ZIF-62 and phosphate glasses are candidates because the melting temperature of ZIF-62 (437 C) is similar to the glass transition temperature of phosphate glass series (31-450 C), which allows for mixing of both materials and thus for making composites. The authors did a lot of efforts to characterize the resulting materials by various techniques and concluded that the composites possess distinct domains of each materials with the size of a few micrometers or more. I believe that the idea of mixing MOFs and inorganic glasses are non-trivial and this manuscript should be potentially accepted in Nature Communications. However, the authors should show the correlation between these new structural feature of mixed glass composites and any bulk properties. One can simply assume how the mechanical property or thermal conductivity can be changed. Otherwise, it is rather difficult to convince the readers to the advantage of making this type of new glass composites. The detail scientific comments are followings;

1. The formation of ZIF-zni by recrystallization would be an interesting topic itself; however, in this logical flow starting from this introduction, it is not really important to do the detail investigation. It would be better to shorten this discussion for clarifying the importance of this manuscript.
2. I did not understand well the part of 'interlocked network structure of pure phase domains'. What does it mean?

Reviewer #2 (Remarks to the Author):

This work describes the formation of mixed inorganic-MOF glass composites by combining the MOF ZIF-62 with different phosphate glasses under heating treatment. The prepared glass materials offered mechanical properties which are intermediate between the native components - this is of potential interest for a broad scope of materials chemists. The opinion of this referee is that this manuscript should be publishable in Nature Communications after major revision.

In my opinion, an important aspect of this work is the occurrence of an interface between the MOF ZIF-62 and the inorganic glasses - which is indeed a very challenging question. This is key to demonstrate the formation of composite glass materials instead of mere physical mixtures, for which one should not expect new properties. The authors used a variety of experimental probes to demonstrate their hypothesis, including, IR, Raman, NMR and PDF analyses. ³¹P NMR data demonstrate the presence of P-N bonds between the phosphate tetrahedra and the Im ligands. Raman spectroscopy showed the occurrence of a weak contribution at 145 cm⁻¹, linked to the formation of Na-N bonds. In addition, the authors have performed PDF analyses to further corroborate the formation of these new bonds locally.

Here come my major concerns about this work:

- Really enjoyed the detailed and careful information included in the SI about the differential PDF analyses. However, I am not completely convinced about the diff-PDF profiles shown in Figures 55-58 in the SI. For the (agZIF-62)_{0.5}(Inorganic Glass)_{0.5} - 1 min samples, one should expect to see the contribution at ca. 1.6 Å linked to the P-N for (agZIF-62)_{0.5}(Al-rich)_{0.5} - 1 min too - in agreement with the ³¹P NMR data. Instead, one can see a negative contribution around similar r values, which could imply overweighting of the initial components. In addition, a positive peak at ca. 1.3 Å (guessing the value) is also observed, which is associated with aromatic C-C correlations of the Im ligands. These two facts together might indicate that the normalization factor could be slightly off. Considering the extremely low concentration of the interface compared to the bulk, it is very difficult to obtain

good diff-PDF signals. As the authors point out: there is the "difficulty of putting X-ray total scattering data on an absolute scale". However, I think it could be worthy to try to re-adjust a bit the normalization factors, which, according the data collection and treatment reported by the authors, should be very close to 1. Another possibility is that, indeed, the concentration of the interface is below the detection limit of the technique.

- Regarding the diff-PDF data corresponding to the (agZIF-62)_{0.5}(Inorganic Glass)_{0.5} – 30 min samples (Figures 57-58) in SI, I have similar concerns. In my opinion, these diff-PDF data look similar to the glass component (although a slight shift to longer r values is observed). This fact might suggest that the glass component is overweighed. Due to the nature of the proposed interface (Figure 9), the new correlations should be clearly different than the ones seen for the inorganic glass.

- Regarding Figure 9, I find the structural model quite misleading. Which are the experimental evidences to propose the formation of Zn-O bonds? Do the authors mean mere interactions? Could not find in the article any explanation about this. Although the authors clarify is only "one possible structure", I am afraid is still confusing. I would suggest to revise or omit it.

- Other minor changes are:

1. Table 1 (Page 5) is redundant, in my opinion. The nomenclature chosen by the authors is clear and self-explanatory.

2. Page 10, PDF section. It reads "Raman and ³¹P NMR data indicated the presence of Na-N and P-N bonding in the composite samples, and so their expected bond distance ranges were measured from crystals of appropriate chemistry as between 2.4 – 3.0 Å ^{35,36} and 1.7 – 1.8 Å, respectively.", while in the Raman section (page 7) one cannot find an interpretation of the band at 145 cm⁻¹. The authors discuss this band later, in page 11, which makes the reading a bit difficult in this part.

3. Table 3-5 in SI: I would recommend to report the NMR chemical shifts with only two decimals instead of three.

4. Figures 37-48 and 55-58 in SI, I would recommend to include values for the most representative PDF contributions (at least below 3 Å).

Reviewer 1.

This manuscript describes the synthesis, characterization and properties of mixed metal-organic framework and inorganic glass composites. Particularly, the authors chose ZIF-62 and phosphate glasses as candidates because the melting temperature of ZIF-62 (437 C) is similar to the glass transition temperature of phosphate glass series (31-450 C), which allows for mixing of both materials and thus for making composites. The authors did a lot of efforts to characterize the resulting materials by various techniques and concluded that the composites possess distinct domains of each materials with the size of a few micrometers or more. I believe that the idea of mixing MOFs and inorganic glasses are non-trivial and this manuscript should be potentially accepted in Nature Communications.

We thank reviewer for their time, and are happy that they consider the manuscript for publication in Nature Communications.

However, the authors should show the correlation between these new structural feature of mixed glass composites and any bulk properties. One can simply assume how the mechanical property or thermal

conductivity can be changed. Otherwise, it is rather difficult to convince the readers to the advantage of making this type of new glass composites. The detail scientific comments are followings;

We agree with the reviewer that, as the materials reported here are extremely novel, there properties are of great interest to the scientific community. As such we have added additional macroscopic measurements: density, scratch-testing and ionic conductivity (**Supplementary Table S10-S12, Figure 5, Supplementary Figure 61-64 and Figure 9**, respectively); these are in addition to the mechanical measurements reported in **Figure 4**. Taken together these measurements establish that these composites display behaviours between those of ZIF glasses and inorganic glasses.

However we would stress that this is the first type of this new material and as the prototypical example of the combination of inorganic glasses with MOF glasses we believe that the characterisation work here provides a basis for other researchers to explore the potential afforded by this new class of composites.

Scratch Resistance Section Added

“The scratch resistance of the $(a_g\text{ZIF-62})_{0.5}(\text{base})_{0.5}$ – 1 min sample, $a_g\text{ZIF-62}$, and the inorganic base glass was also investigated (**Figure 5**). Due to the lower hardness of $a_g\text{ZIF-62}$ ($H(a_g\text{ZIF-62}) = 0.71$ GPa) as compared to the inorganic glass ($H(\text{base}) = 4.47$ GPa) a considerably larger sample volume is deformed during scratching (**Figure 5a**). Despite this mismatch in the indenter displacement (h), very similar values of lateral force (F_L) were recorded for these two glasses (**Figure 5b**), indicating a substantially lower resistance of $a_g\text{ZIF-62}$ against the lateral movement of the indenter tip, i.e. a lower scratch hardness³⁰, as compared to the inorganic base glass. When probing the $(a_g\text{ZIF-62})_{0.5}(\text{base})_{0.5}$ – 1 min glass sample, pronounced fluctuations in both h and F_L are clearly visible during scratching. The length scale of these variations corresponds very well to the microstructural scale of the composite constituents, an effect also observed in the mechanical resistance (i.e. hardness and modulus) of the composite material as revealed by nanoindentation (**Figure 4**). Mean values of h (**Figure 5a**) and F_L (**Figure 5b**) for the $(a_g\text{ZIF-62})_{0.5}(\text{base})_{0.5}$ – 1 min glass sample are in-between the $a_g\text{ZIF-62}$ and inorganic base glasses, which confirms our earlier conclusion that the composite materials are, on average, more compliant than the inorganic base glass but mechanically more stable than pure $a_g\text{ZIF-62}$. The scratch hardness (H_s) represents the work, W_s , which is required to generate (deform) a scratch groove of volume V_s (**Figure 5c**)³⁰, the value of W_s/V_s for the $(a_g\text{ZIF-62})_{0.5}(\text{base})_{0.5}$ – 1 min glass sample is consistently above that of pure $a_g\text{ZIF-62}$ ($H_s(a_g\text{ZIF-62}) = 0.45$ GPa), and close to that of the pure inorganic base glass ($H_s(\text{base}) = 4.84$ GPa).”

Figure 5 Scratch resistance. Spatial variation in a) indenter displacement (h) and b) lateral force (F_L) as monitored during a constant load scratch test across the surface of the $(a_9\text{ZIF-62})_{0.5}(\text{base})_{0.5}$ glass heat treated for 1 minute (red curves). Experimental data from equivalent scratch tests on the inorganic glass (base, green curves) and $a_9\text{ZIF-62}$ (blue curves) are added. Red shades display the confidence intervals (95 %) for a) h and b) F_L as derived from multiple such scratch tests performed on the $(a_9\text{ZIF-62})_{0.5}(\text{base})_{0.5} - 1$ min glass sample. c) Work of deformation (W_s), as derived from the integral of F_L across a distance of $80 \mu\text{m}$ (neglecting the first and last $10 \mu\text{m}$ of each scratch), versus the corresponding scratch volume (V_s) of the investigated glass samples. The scratch hardness (H_s) is defined as the slope of the linear regression curve of W_s versus V_s .

Methods:

Scratch testing. The scratch resistance was analysed in constant-load scratch tests with a three-sided Berkovich diamond tip in edge-forward orientation using the nanoindentation setup mentioned above. The indenter tip was moved across the sample surface along a distance of $100 \mu\text{m}$ at a fixed scratch velocity of $10 \mu\text{m/s}$ and under a prescribed normal load of 10mN , while monitoring the lateral force (F_L) and indenter displacement (h). In total, ten such scratch tests were performed on each sample.

Ionic Conductivity Section Added

"Ionic Conductivity Measurements"

The indication from Raman spectroscopy, that Na^+ ions enter the $a_9\text{ZIF-62}$ domains of the $(a_9\text{ZIF-62})_{0.5}(\text{Inorganic Glass})_{0.5}$ samples, led us to perform ionic conductivity measurements on the $(a_9\text{ZIF-62})_{0.5}(\text{Na-deficient})_{0.5} - 1$ min and 30 min samples. These were carried out in order to demonstrate the nature of the bonding and mobility of sodium ions in the composites, and how both were affected by heat treatment time.

The ionic conductivity was measured between $110\text{--}200 \text{ }^\circ\text{C}$ for the $(a_9\text{ZIF-62})_{0.5}(\text{Na-deficient})_{0.5}$ samples heat treated for 1 minute and 30 minutes, and between $50\text{--}250 \text{ }^\circ\text{C}$ for the pure Na-deficient glass (**Supplementary Figures 61–64**) and the activation energy (E_a) for ion motion was extracted (**Figure 9**). The $a_9\text{ZIF-62} - 1$ min had measurements of less than 10^{-10} S/cm , meaning that an accurate measurement of the conductivity and activation energy could not be obtained due to the lack of mobile ions in the $a_9\text{ZIF-62}$ phase. Therefore in **Figure 9** the $a_9\text{ZIF-62}$ conductivity is reported at the limit of 10^{-10} S/cm , representing an upper bound of conductivity. Although the E_a for sodium conduction in $a_9\text{ZIF-62}$ is unknown, as no Na^+ is present, the conductivity of ionic liquid impregnated amorphous ZIF-8 showed an E_a of approximately 0.3 eV for Na^+ ³⁷, and therefore $a_9\text{ZIF-62}$ is assumed to have a similar

value.

The $(a_g\text{ZIF-62})_{0.5}(\text{Na-deficient})_{0.5} - 1$ min sample showed a reduction in conductivity at 200 °C relative to the Na-deficient glass. This is explained by the addition of the non-conductive $a_g\text{ZIF-62}$ phase which drastically reduces the concentration of sodium ions per volume. This is corroborated by the sharp decrease in densities in the composite (**Supplementary Tables 10–12**). Furthermore the microstructure of the $(a_g\text{ZIF-62})_{0.5}(\text{Na-deficient})_{0.5} - 1$ min sample consists of remnant particles (**Figure 3, Supplementary Figures 8 and 36**), whose interfaces will act as defects reducing conduction. The conductivity of the $(a_g\text{ZIF-62})_{0.5}(\text{Na-deficient})_{0.5} - 30$ min sample is larger, indicating that conductivity increases with annealing time. This is explained by densification (**Supplementary Table 12**) and the grain growth as evident from confocal microscopy and SEM (**Figure 3, Supplementary Figures 8 and 36**), indicating an enhanced $[\text{Na}^+]/\text{cm}^3$ as well as a more efficient sintering, with the latter resulting in a reduction in interfaces and defects.

The activation energy for ion motion is lower in the $(a_g\text{ZIF-62})_{0.5}(\text{Na-deficient})_{0.5} - 1$ min sample than in the Na-deficient sample (i.e. the inorganic glass alone), indicating a low energy pathway for Na^+ motion through the structure. As the Raman data indicates formation of Na-N bonding, the lower E_a could indicate a potential motion of Na^+ ions through the $a_g\text{ZIF-62}$ glass phase, with a lower activation energy due to the phases more porous nature. The experimental error of this measurement however may also be indicative of the large degree of structural heterogeneity observed in this sample by microscopy. However, after extended annealing time, the $(a_g\text{ZIF-62})_{0.5}(\text{Na-deficient})_{0.5} - 30$ min sample shows an activation energy more similar to that of the bulk Na-deficient glass sample and with a reduced experimental error. The increase in E_a may be due to the overall structural densification in which a higher energy but more prevalent conduction pathway through the Na-deficient glass phase predominates over interfacial conduction through $a_g\text{ZIF-62}$ boundaries. Taken together these results indicate that the composite samples show an appreciable degree of conduction of Na^+ ions, whose exact sodium conduction mechanisms are of interest to the active sodium-ion battery community.”

Figure 9 Ionic conductivity measurements. Measurements of ionic conductivity at 200 °C of the $(a_g\text{ZIF-62})_{0.5}(\text{Na-deficient})_{0.5} - 1$, $(a_g\text{ZIF-62})_{0.5}(\text{Na-deficient})_{0.5} - 30$ min and Na-deficient samples along with the activation energy extracted from the gradient of the conductivity-temperature measurements (see methods). Data for $a_g\text{ZIF-62} - 1$ min are predictions only (see text) and are represented with open circles and dashed lines.

Methods Sections Added

“Density

Archimedean Method. The densities of the inorganic glasses were measured 3 – 4 times by the Archimedes principle at RT in absolute ethanol.

Pycnometry. The densities of the crystalline and glassy pure heat-treated ZIF-62, as well as the composites were measured using a Quantachrome Ultrapyc 1200e He pycnometer at 20.0 °C for 5 sets of 30 cycles each. “

“Impedance Spectroscopy

The surface areas and thicknesses of the Na-deficient, $(a_g\text{ZIF-62})_{0.5}(\text{Na-deficient})_{0.5} - 1$ min and $(a_g\text{ZIF-62})_{0.5}(\text{Na-deficient})_{0.5} - 30$ min samples were measured (1 cm² and around 1 mm; roughly 2 mm² and 0.7 mm). All samples were well-polished; the Na-deficient sample was sputtered with a gold layer on both sides, however, the composites were left bare for electrical measurements.

The impedance measurements were performed on a Novocontrol Alpha-A spectrometer paired with a Novotherm Temperature Control System. The measured frequency range was from 10⁻¹ to 10⁷ Hz. The temperatures from 50 to 250 °C with intervals of 25 °C were measured for the Na-deficient sample and for the ZIF samples $(a_g\text{ZIF-62})_{0.5}(\text{Na-deficient})_{0.5} - 1$ min and $(a_g\text{ZIF-62})_{0.5}(\text{Na-deficient})_{0.5} - 30$ min from 50 to 200 °C with intervals of 30 °C

The resistance under direct current (R_{DC}) was determined as the right intersection of the x-axis with the half circle of the Nyquist Plot (real and imaginary part of the impedance, Z' VS Z''), see **Supplementary Figures 61–63**. The conductivity (σ) is calculated as:

$$\sigma = \frac{1}{lR_{DC}A}$$

where l is the thickness and A is the area of the sample. The temperature dependency of the ionic conductivity was described by the Arrhenius relation (see **Supplementary Figure 64**):

$$\sigma T = \sigma_0 \exp\left(-\frac{E_a}{k_B T}\right)$$

where σ_0 is the pre-factor, k_B is the Boltzmann constant and E_a is the activation energy of the ionic conductivity."

Supplementary Data Added

Supplementary Figure 61. Nyquist Plot of $(a_9ZIF-62)_{0.5}(Na-deficient)_{0.5} - 1$ min composite.

Supplementary Figure 62. Nyquist Plot of $(\text{agZIF-62})_{0.5}(\text{Na-deficient})_{0.5}$ – 30 min composite.

Supplementary Figure 63. Nyquist Plot of Na-deficient inorganic sample.

Supplementary Figure 64. Arrhenius plots of $(a_g\text{ZIF-62})_{0.5}(\text{Na-deficient})_{0.5}$ – 1 min, $(a_g\text{ZIF-62})_{0.5}(\text{Na-deficient})_{0.5}$ – 30 min and Na-deficient inorganic samples.

Supplementary Table 10. Densities (ρ), as measured by the Archimedeian method, of the $(1-x)([\text{Na}_2\text{O}]_x[\text{P}_2\text{O}_5])_x-x([\text{AlO}_{3/2}][\text{AlF}_3]_y)$ inorganic glass series. Error was calculated at 95% confidence level.

Sample	Density (g/cm^3)	Error (g/cm^3)
base	2.64	0.01
Na-deficient	2.75	0.02
Al-rich	2.71	0.01

Supplementary Table 11. Densities (ρ), as measured by pycnometry, of the crystalline and amorphous pure ZIF-62 controls (1 and 30 min heat treatments). Error was calculated at 95% confidence level.

Sample	Density (g/cm^3)	Error (g/cm^3)
ZIF-62	1.47	0.08
$a_g\text{ZIF-62}$ – 1 min	1.38	0.09
$a_g\text{ZIF-62}$ – 30 min	1.42	0.04

Supplementary Table 12. Densities (ρ), as measured by pycnometry, of composites heat treated for 1 and 30 min. Error was calculated at 95% confidence level.

Sample	1 min		30 min	
	Density (g/cm^3)	Error (g/cm^3)	Density (g/cm^3)	Error (g/cm^3)
$(a_g\text{ZIF-62})_{0.5}(\text{base})_{0.5}$	1.60	0.1	1.84	0.3
$(a_g\text{ZIF-62})_{0.5}(\text{Na-deficient})_{0.5}$	1.69	0.3	1.83	0.1
$(a_g\text{ZIF-62})_{0.5}(\text{Al-rich})_{0.5}$	1.72	0.06	1.82	0.04

1. The formation of ZIF-zni by recrystallization would be an interesting topic itself; however, in this logical flow starting from this introduction, it is not really important to do the detail investigation. It would be better to shorten this discussion for clarifying the importance of this manuscript.

We agree with the reviewer that there is a large amount of information in this manuscript, and so to follow his/her advice to improve the flow and readability of this manuscript we have shortened the section on ZIF-zni formation in the discussion. The text that formally read:

"The X-ray diffraction experiments performed on a sample of $(a_g\text{ZIF-62})_{0.5}(\text{base})_{0.5} - 1$ min indicate recrystallization to ZIF-zni; small faceted crystals of which were also visible by SEM (**Figure 8**)³². However, continued heating resulted in disappearance of these peaks, the $(a_g\text{ZIF-62})_{0.5}(\text{base})_{0.5} - 30$ min sample was completely amorphous. ZIF-zni is a dense, mono-linker imidazolate ZIF $[\text{Zn}(\text{Im})_2]$ that is reported in the literature as forming when ZIF-4 is heated to approx. 370 °C before melting at approx. 590 °C^{21,40}. ZIF-62 has not previously been reported to recrystallise on heating, with the absence of recrystallisation being explained by the bulkier benzimidazolate linker imposing added steric constraints on the ZnN_4 coordination polyhedra²¹. Apart from the presence of the bulkier benzimidazole ligand in ZIF-62, ZIF-4 and ZIF-62 have closely interrelated structures: both crystallise in the *Pbca* space group with the *cag* topology^{41,42}. Previous work has also shown that benzimidazole and zinc metaphosphate glass are miscible and react to form a clear solid material after heating at 160 °C for 24 hours⁴³.

Therefore, supported by experimental results and consistent with the literature, the following is proposed to explain the observed formation of ZIF-zni:

1. Intimate mixing occurs between the inorganic glass and ZIF-62 upon heating and isothermal heat treatment above $T_m(\text{ZIF-62})$ and $T_g(\text{inorg})$.
2. This causes the depletion of benzimidazolate ligands in the ZIF-glass, leaving behind a defect-rich $\text{Zn}(\text{Im})_2$ framework at the interface between the inorganic and ZIF liquids.
3. This imidazolate-rich region at the interface resembles a ZIF-4 structure, and thus undergoes recrystallisation to ZIF-zni.
4. The liquid ZIF promotes melting of the newly formed ZIF-zni phase⁴⁴; and so ZIF-zni melts on further heating.

In an apparent contrast, both $(a_g\text{ZIF-62})_{0.5}(\text{Na-deficient})_{0.5} - 1$ min and $(a_g\text{ZIF-62})_{0.5}(\text{Al-rich})_{0.5} - 1$ min samples were completely amorphous, though some recrystallization to the dense ZIF-zni ($\text{Zn}(\text{Im})_2$) phase was observed in the corresponding samples heated for 30 minutes (**Figure 8c inset**). However, the inorganic glass in both these samples is expected to be more viscous due to higher glass transition temperatures. The delayed onset of ZIF-zni formation is therefore explained by the sluggish liquid mixing of the inorganic and ZIF components, which is necessary for reaction to occur."

Has been shortened considerably and replaced with:

"The X-ray diffraction and SEM experiments performed on a sample of $(a_g\text{ZIF-62})_{0.5}(\text{base})_{0.5} - 1$ min indicate a small degree of recrystallization to the dense $[\text{Zn}(\text{Im})_2]$ polymorph, ZIF-zni (**Supplementary Figure 65**)³⁴. Continued isothermal treatment results in subsequent reduction of the ZIF-zni phase in the $(a_g\text{ZIF-62})_{0.5}(\text{base})_{0.5} - 30$ min sample. This reduction in Bragg scattering was further confirmed by PDF with the $S(Q)^{\text{Diff}}$ confirming that sharp Bragg features were still present in $(a_g\text{ZIF-62})_{0.5}(\text{base})_{0.5} - 30$ min, though to a smaller degree. The $(a_g\text{ZIF-62})_{0.5}(\text{Al-rich})_{0.5} - 30$ min and $(a_g\text{ZIF-62})_{0.5}(\text{Na-deficient})_{0.5} - 30$ min samples contained Bragg peaks ascribed to ZIF-zni (**Supplementary Figure 65**),

though in contrast, the $(a_9\text{ZIF-62})_{0.5}(\text{Al-rich})_{0.5} - 1$ min and $(a_9\text{ZIF-62})_{0.5}(\text{Na-deficient})_{0.5} - 1$ min did not.

ZIF-zni is reported to recrystallise from ZIF-4, a $\text{Zn}(\text{Im})_2$ polymorph sharing the same cag topology as ZIF-62^{39,40}, on heating to approx. 370 °C before melting at approx. 590 °C^{21,41}. The absence of recrystallisation in ZIF-62 has been ascribed to the bulkier benzimidazolate linker imposing added steric constraints on the ZnN_4 coordination polyhedra²¹. We therefore postulate that ZIF-zni formation arises in this case due to an interaction between the inorganic and MOF glass phases, with stronger interactions occurring at lower viscosities of the inorganic glass component. This may proceed via migration of benzimidazole to the inorganic glass, which is consistent with prior literature showing that benzimidazole and zinc metaphosphate glass are miscible⁴²; recrystallisation to ZIF-zni of the remnant Im-rich interface domains then occurs, before this itself either melts, or is dissolved by the melt on further heating in the base 30 minute sample. The effect is most pronounced in those samples with lower glass transition temperatures and hence lower viscosities at the treatment temperatures, which promote a greater degree of mixing.”

We have moved figure 8 from the main paper to the supplementary information, it is now marked as Supplementary Figure 65.

2. I did not understand well the part of 'interlocked network structure of pure phase domains'. What does it mean?

We intended to convey that the composites seem to have separate regions of predominantly ZIF-62 or inorganic glass origin, bonded at the interfaces, as identified by EDX and SEM analysis, see supplementary figures 28-33. However we have clarified this in the main text which now reads:

"These results describe a new class of inorganic – MOF glass composites, prepared by heating a mixture of a phosphate glass and ZIF-62. The composites formed upon cooling contain two distinct glass transition temperatures, matching those of $a_9\text{ZIF-62}$ and the relevant inorganic glass, implying that the composite contains separate domains of each glass phase bonded at their interface into a single solid body in agreement with SEM, mechanical and conductivity results. "

Reviewer 2.

This work describes the formation of mixed inorganic-MOF glass composites by combining the MOF ZIF-62 with different phosphate glasses under heating treatment. The prepared glass materials offered mechanical properties which are intermediate between the native components - this is of potential interest for a broad scope of materials chemists. The opinion of this referee is that this manuscript should be publishable in Nature Communications after major revision.

We thank the reviewer for their time, and we are pleased to find that they agree with reviewer 1 in their assessment that this manuscript is publishable in Nature Communications.

In my opinion, an important aspect of this work is the occurrence of an interface between the MOF ZIF-62 and the inorganic glasses – which is indeed a very challenging question. This is key to demonstrate the formation of composite glass materials instead of mere physical mixtures, for which one should not expect new properties. The authors used a variety of experimental probes to demonstrate their hypothesis, including, IR, Raman, NMR and PDF analyses. ^{31}P NMR data demonstrate the presence of P-N bonds between the phosphate tetrahedra and the Im ligands. Raman spectroscopy showed the occurrence of a weak contribution at 145 cm^{-1} , linked to the formation of Na-N bonds. In addition, the authors have performed PDF analyses to further corroborate the formation of these new bonds locally.

The reviewer has accurately picked out one of the most exciting and challenging aspects of this work, which was the characterisation of the interaction between the MOF phase and inorganic glass phases at the interface.

Here come my major concerns about this work:- Really enjoyed the detailed and careful information included in the SI about the differential PDF analyses.

Once again the authors would like to thank the reviewer for their positive comments and very detailed reading of this section.

However, I am not completely convinced about the diff-PDF profiles shown in Figures 55-58 in the SI. For the $(\text{agZIF-62})_{0.5}(\text{Inorganic Glass})_{0.5}$ – 1 min samples, one should expect to see the contribution at ca. 1.6 \AA linked to the P-N for $(\text{agZIF-62})_{0.5}(\text{Al-rich})_{0.5}$ – 1 min too - in agreement with the ^{31}P NMR data. Instead, one can see a negative contribution around similar r values, which could imply overweighting of the initial components. In addition, a positive peak at ca. 1.3 \AA (guessing the value) is also observed, which is associated with aromatic C-C correlations of the Im ligands. These two facts together might indicate that the normalization factor could be slightly off. Considering the extremely low concentration of the interface compared to the bulk, it is very difficult to obtain good diff-PDF signals. As the authors point out: there is the "difficulty of putting X-ray total scattering data on an absolute scale". However, I think it could be worthy to try to re-adjust a bit the normalization factors, which, according to the data collection and treatment reported by the authors, should be very close to 1.

This a very astute point and detailed feedback. We acknowledge that the difficulty with a differential technique such as this, is that the resulting diffD(r) are relatively sensitive to changes in normalisation. The data presented here represents our best attempts to normalise such that positive/negative peaks in the diffD(r), which have been identified as belonging to either the inorganic glass or a_gZIF-62, are minimised. We have gone to considerable lengths to determine the normalisation factors by the process described in the supplementary information (see **Table 1**) which, in agreement with the comment by the reviewer, are all very close to 1. The protocol we used was to weight the higher-Q data more strongly than the lower-Q data, since the levels in the high-Q data are governed by the amount of material from each sample independent of the structure of each material. Hence large differences in the low-Q scattering will not necessarily be minimised, as seen for example when a_gZIF-62 transforms to ZIF-zni (SI Figure 52), although the low-Q differences will not overly influence the low-r features in the PDFs. We are reassured that the differences are small when a_gZIF-62 remains amorphous (see, for example, SI Figure 51) and that the fitted scale factors are all close to 1 and do not change very much between the different mixtures. We also note that the oscillations in the PDFs are weak, except for when the ZIF-62 component transforms from glassy to crystalline character.

Table 1: Scale factors used in the paper (equation 5, supplementary information) introduced into the differential PDF method to accommodate for the inability of X-ray Total-Scattering to be placed on an absolute scale.

Sample	a _g ZIF-62 Scale	Inorganic Scale	Sample Scale
(a _g ZIF-62) _{0.5} (Na-deficient) _{0.5} – 1 min	0.90	1.08	1.02
(a _g ZIF-62) _{0.5} (Na-deficient) _{0.5} – 30 min	0.90	1.08	1.02
(a _g ZIF-62) _{0.5} (Al-rich) _{0.5} – 1 min	0.85	1.17	1.04
(a _g ZIF-62) _{0.5} (Al-rich) _{0.5} – 30 min	0.85	1.17	1.04
(a _g ZIF-62) _{0.5} (Base) _{0.5} – 1 min	0.85	1.17	1.01
(a _g ZIF-62) _{0.5} (Base) _{0.5} – 30 min	0.85	1.17	1.01

Another possibility is that, indeed, the concentration of the interface is below the detection limit of the technique.

- Regarding the diff-PDF data corresponding to the (a_gZIF-62)_{0.5}(Inorganic Glass)_{0.5} – 30 min samples (Figures 57-58) in SI, I have similar concerns. In my opinion, these diff-PDF data look similar to the glass component (although a slight shift to longer r values is observed). This fact might suggest that the glass component is overweighed. Due to the nature of the proposed interface (Figure 9), the

new correlations should be clearly different than the ones seen for the inorganic glass.

We agree with the reviewer that the picture in the short-range order region (1-6 Å) remains relatively opaque. As such we have changed the text substantially in the manuscript; we show that the largest difficulty with this analysis is that the ZIF component is changing during the process (the $S(Q)^{\text{Diff}}$ in the Inset to Figure 8a shows this very clearly); we now state that the fit is weighted to give the best agreement at high-Q (the level of which reflects the sample composition the best – independent of structure of the samples – and to produce truncation-ripple free diff-PDFs); and we estimate that the interface structural changes are smaller than the changes in the ZIF-62 sample itself.

To take into account the reviewers critique we have amended the PDF section to remove commentary on new correlations which previously read:

“The $D(r)^{\text{Diff}}$ of all the samples contain features due to the inorganic glass and ZIF-62 because of the difficulty of putting X-ray total scattering data on an absolute scale (**See Supplementary Methods**). In spite of this, a tentative attempt is made to assign the new correlations observed in the $D(r)^{\text{Diff}}$. Raman and ^{31}P NMR data indicated the presence of Na-N and P-N bonding in the composite samples, and so their expected bond distance ranges were measured from crystals of appropriate chemistry as between 2.4 – 3.0 Å^{35,36} and 1.7 – 1.8 Å^{37,38} respectively. It can be seen that there is a peak at approximately 2.6-2.7 Å consistent with Na-N in all $(\text{a}_g\text{ZIF-62})_{0.5}(\text{Inorganic Glass})_{0.5}$ – 1 min samples. In the $(\text{a}_g\text{ZIF-62})_{0.5}(\text{Inorganic Glass})_{0.5}$ – 30 min samples and in the $(\text{a}_g\text{ZIF-62})_{0.5}(\text{base})_{0.5}$ – 1 min sample a peak at approximately 1.6 Å is observed, which is broadly in the range expected for a P-N correlation. However the peak assigned as Na-N in the $(\text{a}_g\text{ZIF-62})_{0.5}(\text{Inorganic Glass})_{0.5}$ – 30 min samples decreased in length to between 2.5-2.6 Å, which is in better agreement with an expected correlation from the pure inorganic glass (**Supplementary Figure 55 and 57**).”

Has now been replaced with:

" Real space data, $D(r)^{\text{Diff}}$, were obtained by Fourier transform of the structure factor, $S(Q)^{\text{Diff}}$, corresponding to these intensity differences (**Supplementary Figure 55-60**). However, the $D(r)^{\text{Diff}}$ of all the samples contain residual features due to the inorganic glass, and/or ZIF-62 or ZIF-zni $D(r)$ s. Moreover, no correlations that could be definitively ascribed to new bonds observed through Raman scattering or ^{31}P NMR data could be observed (**Supplementary Figure 57 and 59**). These observations are explained by the unexpected change in the nature of the ZIF component upon heating (**Figure 8a insert, Supplementary Figure 55 and 56**), alongside the low interfacial interaction volume, meaning that new correlations may be below the detectable limit of the technique. The $D(r)^{\text{Diff}}$ of the $(\text{a}_g\text{ZIF-62})_{0.5}(\text{Inorganic Glass})_{0.5}$ – 30 min and the $(\text{a}_g\text{ZIF-62})_{0.5}(\text{base})_{0.5}$ – 1 min are however all qualitatively similar as expected from the similar Bragg scattering observed in the $S(Q)^{\text{Diff}}$ (**Supplementary Figure 57-60**), which is attributed to the formation of ZIF-zni in the heat treated composite samples.

Long-range order (LRO) was also evident in the $D(r)^{\text{Diff}}$ from $(\text{a}_g\text{ZIF-62})_{0.5}(\text{base})_{0.5}$ – 1 min, $(\text{a}_g\text{ZIF-62})_{0.5}(\text{Na-deficient})_{0.5}$ – 30 min and $(\text{a}_g\text{ZIF-62})_{0.5}(\text{Al-rich})_{0.5}$ – 30 min samples. However the $D(r)^{\text{Diff}}$ $(\text{a}_g\text{ZIF-62})_{0.5}(\text{Base})_{0.5}$ – 30 min sample appeared flat at extended distances, which is due to the very

small proportion of crystalline component as seen in the very small Bragg features in the $S(Q)_{\text{Diff}}$ (Supplementary Figure 58 and 60).”

Regarding Figure 9, I find the structural model quite misleading. Which are the experimental evidences to propose the formation of Zn-O bonds? Do the authors mean mere interactions? Could not find in the article any explanation about this. Although the authors clarify is only "one possible structure", I am afraid is still confusing. I would suggest to revise or omit it.

There is direct experimental evidence from ^{31}P NMR for P-N bonds and from Raman for Na-N bonds. The reviewer is correct that there is no direct measure of Zn-O-P correlations, however there existence was included due to the necessity of charge balance and the likelihood of Zn^{2+} remaining fully coordinated. We have amended the manuscript to make this clear:

“Zn-O-P correlations though not directly experimentally measured were included for reasons of charge balance and to maintain tetrahedral coordination of Zn centres, there inclusion is also justified by the

large number of examples of inorganic glasses which contain similar structures^{43–45}.” -

Other minor changes are:

1. Table 1 (Page 5) is redundant, in my opinion. The nomenclature chosen by the authors is clear and self-explanatory.

We have removed table 1 from the manuscript.

2. Page 10, PDF section. It reads "Raman and ^{31}P NMR data indicated the presence of Na-N and P-N bonding in the composite samples, and so their expected bond distance ranges where measured from crystals of appropriate chemistry as between 2.4 – 3.0 Å^{35,36} and 1.7 – 1.8 Å, respectively.", while in the Raman section (page 7) one cannot find an interpretation of the band at 145 cm^{-1} . The authors discuss this band later, in page 11, which makes the reading a bit difficult in this part.

We have included the interpretation of the 145 cm^{-1} band (assigned to Na-N) in the ‘Structural Investigations’ section to improve the readability of the manuscript:

"We link the reaction to the formation of new Na—N bonds, given similar peaks in sodium imidazolate-containing compounds at 161 and 136 cm^{-1} ³³, and an as-purchased pure compound sodium imidazolidine derivative (strong peak at 150 cm^{-1}) (Supplementary Figure 26)."

3. Table 3-5 in SI: I would recommend to report the NMR chemical shifts with only two decimals instead of three.

We have amended the tables to be to two decimal places.

4. Figures 37-48 and 55-58 in SI, I would recommend to include values for the most representative PDF contributions (at least below 3 Å).

We have included the values of the peak positions in the SRO region of the PDF in accordance with the wishes of the reviewer.

Yours sincerely,

Thomas Bennett

Dr. Thomas Douglas Bennett

REVIEWERS' COMMENTS

Reviewer #1 (Remarks to the Author):

In the revised manuscript, the authors gave a lot of efforts to clarify the importance of this new type of composites between MOFs and inorganic glasses. The authors not only showed detailed characterizations to understand their composite structures and the interfaces, but demonstrated the mechanical and ion conductive properties of the resulting composites, which were intermediate between their parent materials of MOFs and inorganic glass, which makes this manuscript more attractive to general readers in the field of materials science besides the particular MOFs and glass fields. Therefore, I am happy to suggest the acceptance of this manuscript in Nature Communications as is.

Reviewer #2 (Remarks to the Author):

The authors have addressed in detail all my comments and concerns about this work. My opinion is that now this version should be accepted in Nature Communications.